# INTRINSIC USER-CENTRIC INTERPRETABILITY THROUGH GLOBAL MIXTURE OF EXPERTS

**Vinitra Swamy**
EPFL
vinitra.swamy@epfl.ch

**Syrielle Montariol**
EPFL
syrielle.montariol@epfl.ch

**Julian Blackwell**
EPFL
julian.blackwell@epfl.ch

**Jibril Frej**
EPFL
jibril.frej@epfl.ch

**Martin Jaggi**
EPFL
martin.jaggi@epfl.ch

**Tanja Käser**
EPFL
tanja.kaeser@epfl.ch

## ABSTRACT

In human-centric settings like education or healthcare, model accuracy and model explainability are key factors for user adoption. Towards these two goals, intrinsically interpretable deep learning models have gained popularity, focusing on accurate predictions alongside faithful explanations. However, there exists a gap in the human-centeredness of these approaches, which often produce nuanced and complex explanations that are not easily actionable for downstream users. We present `InterpretCC` (interpretable conditional computation), a family of intrinsically interpretable neural networks at a unique point in the design space that optimizes for ease of human understanding and explanation faithfulness, while maintaining comparable performance to state-of-the-art models. `InterpretCC` achieves this through adaptive sparse activation of features before prediction, allowing the model to use a different, minimal set of features for each instance. We extend this idea into an interpretable, global mixture-of-experts (MoE) model that allows users to specify topics of interest, discretely separates the feature space for each data point into topical subnetworks, and adaptively and sparsely activates these topical subnetworks for prediction. We apply `InterpretCC` for text, time series and tabular data across several real-world datasets, demonstrating comparable performance with non-interpretable baselines and outperforming intrinsically interpretable baselines. Through a user study involving 56 teachers, `InterpretCC` explanations are found to have higher actionability and usefulness over other intrinsically interpretable approaches.

## 1 INTRODUCTION

The rise in popularity of neural networks (NNs) over traditionally interpretable models has come with a severe weakness: the lack of transparency of their predictions. Neural networks are considered black-box models due to their high number of parameters and complex operations (Molnar, 2020). Humans cannot understand how neural network decisions are made under the hood; this is a crucial limitation in sensitive applications such as education or healthcare, where models' predictions might impact humans (Conati et al., 2018; Vellido, 2020).

Towards transparency for neural networks, a range of explainable AI methods have emerged across two main axes: *global* vs. *local* explanations, and *post-hoc* vs. *intrinsic* explanations (Du et al., 2019). *Global* interpretability allows users to understand how the entire model operates by examining its structure and parameters, while *local* interpretability focuses on understanding the rationale behind an individual prediction (Molnar, 2020). Most methods are *post-hoc*, where the explanation is extracted from a model that has already been trained. In contrast, *intrinsic* methods directly incorporate interpretability into the model's structure.

Popular post-hoc methods include attribution methods like LIME (Ribeiro et al., 2016) or SHAP (Lundberg and Lee, 2017), counterfactual methods like CEM (Dhurandhar et al., 2018), or pattern-based methods like PREMISE (Hedderich et al., 2022). Post-hoc methods require users to trust the explainer's approximation of the ground truth explanation (the underlying model's decision process), although they have been shown to be systematically biased and inconsistent (Krishna et al.,

Figure 1: **InterpretCC Architectures**: *Feature Gating (left, individual features)*: (i) All features are input into a discriminator network that outputs a sparse feature activation mask; (ii) Only the features selected via the mask are passed to a predictive network for the final prediction. *Group Routing (right, pre-defined feature groups)*: (i) Features are statically assigned to distinct groups, with each feature routed to only one group; (ii) Features are input to a discriminator network, generating a sparse group activation mask; (iii) Predictions from activated sub-networks (selected via mask) are aggregated by a weighted sum to produce the final output.

2022; Swamy et al., 2022b). *In-hoc* explanations are a subset of post-hoc methods that require access to model weights instead of treating the model as a black box (Molnar, 2020). For instance, Grad-CAM (Selvaraju et al., 2019) assesses the contribution of a component to the model's output, and TCAV/DTCAV (Kim et al., 2018; Ghorbani et al., 2019a) use user-defined concepts to interpret neural network embeddings.

Recent intrinsically explainable model literature has focused on example-based approaches, overwhelmingly for the image modality (*e.g.* B-cos networks (Böhle et al., 2022), PIP-Net (Nauta et al., 2023), ProtoPNet (Chen et al., 2019)) and less commonly in time-series, tabular, or text modalities (Sawada and Nakamura, 2022). Other approaches like NAM (Agarwal et al., 2021) and EBM (Nori et al., 2019) train a model for each input feature or combination of features and output predictions using scores from these models, requiring a lot of subnetworks when the feature space is large. Research towards interpretable mixture-of-experts models has highlighted a hierarchical neural network structure with subnetworks, combining interpretable experts (*i.e.,* decision trees) with NNs for partially interpretable points (Ismail et al., 2023), selectively activating experts (Li et al., 2022), or extracting automated concepts over the input space for routing (You et al., 2023; Alvarez Melis and Jaakkola, 2018). All of these approaches, while expressive, are burdened with overly detailed explanations that limit human understandability and actionability.

In this paper, we therefore present a **user-centric intrinsically interpretable** framework that achieves faithful local interpretability and provides sparse and actionable human-centric explanations, while maintaining comparable predictive performance to its black-box counterparts. To achieve these goals, we use conditional computation to craft interpretable neural pathways using two different architectures (see Figure 1) based on routing through individual features or entire feature groups.

Our models enable statements like the following: "The student's regularity and video watching behavior were the only two aspects selected as important for the student's prediction of passing the course, and the model did not use any other aspects to make this prediction". We refer to interpretability from the users' perspective, focusing on the model's local reasoning for a decision on a specific data point, as opposed to a global understanding of the model's internals. Our models are characterized by sparse explicit routing, truncated feature spaces, and adaptivity per data point. These traits are important for human-centric trustworthiness as they provide clear and concise instance-level explanations (Miller, 2019; Swamy et al., 2023b).

With our family of `InterpretCC` models, we provide the following contributions:

[1] **`InterpretCC` Feature Gating: A simple, interpretable NN architecture** using a gating mechanism to sparsely activate specific features.
[2] **`InterpretCC` Group Routing: An interpretable mixture-of-experts architecture** that uses human-specified group routing to separate the feature space and sparsely activate specific experts.
[3] **An extension of intrinsic interpretability to multiple human-centric modalities and domains**, focusing on time-series (education), tabular (health, synthetic data), and text (sentiment, news).[1]

---

[1]We do not focus on vision datasets since extracting concepts from vision has been well-studied by existing, modality-specific interpretability approaches, *e.g.* Böhle et al. (2022); You et al. (2023); Donnelly et al. (2022); Thomas et al. (2023).

**[4] A novel XAI user study** comparing teachers' preferences of interpretable-by-design model explanations towards designing educational interventions for struggling students.

Across experiments on eight diverse datasets, we show that `InterpretCC` models perform comparatively to non-interpretable baselines (matching or exceeding 95% CIs in 15 of 16 comparisons) and outperform intrinsically interpretable baselines (`ICC` Feature Gating is 9.05% better on average than SENN Features and 3.27% better on average than NAM, while `ICC` Group Routing is on average 5.63% better than SENN Concepts). Moreover, participants of the user study preferred `ICC` explanations over baselines in terms of actionability, usefulness, conciseness and trustworthiness. We provide our code open source: `https://github.com/epfl-ml4ed/interpretcc`.

## 2 BACKGROUND

**Architecture Foundations.** *Conditional Computation (CC)* has become widely used to improve the computationally expensive training and inference of large neural networks by activating only parts of the network (Bengio et al., 2013; 2016; Davis and Arel, 2013). Inspired by the foundations laid out by CC, mixture-of-expert models have rapidly gained popularity for improving the efficiency of neural networks through activating different expert subnetworks at different layers. BASE layers (Lewis et al., 2021) direct each token to a designated expert and Switch Transformers (Fedus et al., 2022b) use CC to select one out of 4 feedforward networks across each transformer layer, optimizing computational resources. Mixtral (Jiang et al., 2024) is a recent LLM using a mixture of experts to select 2 out of 8 expert networks at each layer, reducing the numbers of active parameters by a factor of 4 compared to training, while allowing each token to have access to all the parameters.

| Method | Explanation | | | | |
| | *Granularity* | *Basis* | *Faithfulness* | *Sparsity* | *Stage* |
|---|---|---|---|---|---|
| **LIME, SHAP** (Post-Hoc) | Feature | Use all input features | Approximation | Sparse | Explanation not used in model |
| **TCAV** (In-Hoc) | Concept | User defines concepts through examples | Aligned with concepts | Sparse | Explain from model internals |
| **SENN** | Feature | Use all input features | Guaranteed | Not sparse | Explain then predict |
| | Concept | Automated concept selection | Aligned with concepts | Not sparse | Explain then predict |
| **NAM** | Feature | Use all input features | Guaranteed | Not sparse | Explain then predict |
| **InterpretCC** (Feature Gating, Group Routing) | Feature | Use all input features | Guaranteed | Sparse | Explain then predict |
| | Concept | User (or LLM) defines groups of features | Guaranteed | Sparse | Explain then predict |

Table 1: **Design Comparison**: `InterpretCC` models are at the unique intersection of flexible explanation *granularity* (either features or concepts), guaranteed explanation *faithfulness* to the model's decision process, optimal *sparsity* in the explanation, and explanations used in the prediction (*stage*). *Basis* describes the foundation of the explanation (*e.g.,* user-defined concepts or raw features). A taxonomy can be found in Appendix A.

With `InterpretCC`, we extend a similar routing idea with instance-dependent gating decisions towards an interpretability objective as opposed to only an efficiency or performance objective.

**Interpretability Foundations.** Explainability can be integrated into different stages of the modeling pipeline: post-hoc (after model training), in-hoc (requiring model weights), and intrinsic (interpretable by design) (Swamy et al., 2023b; Molnar, 2020). Our positioning of the design of `InterpretCC` in comparison to popular approaches is described in Table 1. Specifically, we categorize the approaches using their basis and the stage they are applied to in the pipeline as well as four key aspects for human-centric explanations: *faithfulness*, the explanation reflects the model behavior with certainty (Lyu et al., 2024; Dasgupta et al., 2022); *sparsity*, the model uses a minimal amount of features, optimizing for user understandability and actionability (Sun et al., 2024; Ayoobi et al., 2023); *predictive stage*, the explanation covers the entirety of what the model uses for prediction (Speith, 2022; Schwalbe and Finzel, 2024); and *granularity*, the explanation is conveyed in terms of features or concepts (Miller, 2019; Jain et al., 2020).

Post-hoc methods such as LIME (Ribeiro et al., 2016) or SHAP (Lundberg and Lee, 2017) approximate what the model finds important, and therefore cannot be considered faithful (cross-feature actions are often not described in the explanation). LIME and SHAP use the full input feature space and can be configured for sparseness, although they have been shown to choose a broad amount of features (Swamy et al., 2022b) using default settings. In-hoc interpretability approaches often require users to specify examples to define human-understandable concepts (*e.g.* TCAV (Kim et al., 2018), DTCAV (Ghorbani et al., 2019a)), or use hybrid methods with both human-defined and automated concepts (Sawada and Nakamura, 2022). In-hoc approaches have limited faithfulness due to the (lack of) completeness of the concepts and do not use explanations directly in the prediction. Inspired by these approaches, `InterpretCC` allows users to specify interpretable concepts that are directly useful to them. However, we do not use examples, but instead allow users to specify a grouping over the feature space, achieving both sparsity and explanations used for prediction.

Initial approaches have explored expert models for intrinsic interpretability. The Interpretable Mixture of Experts (IME) framework (Ismail et al., 2023) uses linear models alongside deep models to provide partially faithful explanations. LIMoE (Mustafa et al., 2022) focuses on visual experts to identify concepts like textures and faces, enhancing interpretability in vision tasks. Similarly, the Sum-of-Parts (SOP) model (You et al., 2023) uses sparse feature groups to emphasize the model's reliance on subsets of features for predictions. Approaches in extractive rationale methods and explain-then-predict methods (Jain et al., 2020; Bastings et al., 2019; Yu et al., 2019) produce intuitive text explanation guarantees with explanation selection before prediction, but are often not generalizable beyond that modality (see Appendix I.4). Few intrinsic approaches use expert knowledge to define concepts directly, instead using prototype examples (Koh et al., 2020) or rules (Konstantinov and Utkin, 2024). The most relevant models to our work are Self-Explaining Neural Networks (SENN) (Alvarez Melis and Jaakkola, 2018) and Neural Additive Models (NAM) (Agarwal et al., 2021), both neural models similar to `ICC` as opposed to Explainable Boosted Machines (EBM) which uses trees (Nori et al., 2019). SENN extracts concepts with prototypical examples and their relevances, but it lacks faithfulness (it cannot explain what is not in a concept), sparsity (it explains all concepts), and concepts do not cover the entire feature space. NAM assigns a model to each feature and combines the outputs linearly, achieving faithfulness but not sparsity, as all features contribute. `ICC` differs by filtering the feature space instead of using all features (sparsity), using user-defined concepts instead of automated concepts (basis), and assigning each feature to a single group, making feature use explicit (predictive stage).

## 3 METHODOLOGY

Given an input $x$, the objective of our approach is to select a sparse subset of $x$ that will be used to predict the output to solve the classification task. We propose two architectures:

**Feature Gating:** The approach only processes a subset of the features by applying a sparse mask $\mathcal{M}$ on the input $x$ before processing it by a model $f$. The output is given by: $f(\mathcal{M}(x))$.

**Group Routing:** A sparse mixture of models (Fedus et al., 2022a) applied on human-interpretable groups of features where each expert is assigned to a group of features: $\sum_{i=1}^{K} G(x)_i \cdot f_i(\mathcal{M}_{\mathcal{G}}(x)_i)$ where $\mathcal{M}_{\mathcal{G}}(.)_i$ is a sparse mask selecting only the features of group $i$, $f_i$ is the expert model associated with the $i$-th group, and $G(x)_i$ is the output of the gating network for group $i$. If $G(x)_i = 0$, the entire group of features is ignored.

### 3.1 FEATURE GATING

`InterpretCC` Feature Gating, shown in Fig. 1, is the first step towards using CC paths for interpretability. The features are first passed through a discriminator network $D$ to select which ones to use for computing the output. The Gumbel Softmax trick (Jang et al., 2017) is applied on each dimension of $D(x)$ to select features in a differentiable way (see Appendix G for more details). A feature $j$ is activated (the associated value in the mask is non-zero) if the Gumbel Softmax output exceeds a threshold $\tau$, a hyperparameter. This allows the model to adaptively select the number of features based on each instance, using fewer features for simpler cases and more for complex ones.

The output is computed using a model $f$ on the masked input $\mathcal{M}(x)$. Since the explainability is at the feature level, using a black box model for $f$ does not detract from the interpretability. Notably, `ICC` FG does not require human specification.

### 3.2 GROUP ROUTING

We build upon the instance-dependent gating architecture with feature groups. As displayed in Figure 1, instead of selecting features individually, the mask is applied to human interpretable groups of features. Doing so encourages cross-feature interactions while maintaining a meaningful grouping for human users. To select the features belonging to group $i$, we use a binary mask $\mathcal{M}_{\mathcal{G}}(x)_i$ that is computed using human-specified rules. In section 4, we detail our approach to compute $\mathcal{M}_{\mathcal{G}}(x)_i$ for each dataset used in our experiments.

`InterpretCC` Group Routing is a sparse mixture of experts utilizing a gating network to assign a weight $G(x)_i$ to each group. This process mirrors that of Feature Gating, starting with a discriminator network $D_{\mathcal{G}}$ with an input of all features and output of $K$ dimensions ($K$ is the number of groups). It then applies the Gumbel Softmax and a threshold $\tau_{\mathcal{G}}$ to each group. The model output is a weighted sum of the output of each expert $f_i$ that only uses the features from the $i$-th group as input. Using

| | Dataset | Non-interpretable Base Module | Feature-Based Interpretability | | | | Concept-Based Interpretability | | |
|---|---|---|---|---|---|---|---|---|---|
| | | | NAM | SENN Features | FRESH | InterpretCC Feature Gating | SENN Concepts | InterpretCC Top K Routing | InterpretCC Group Routing |
| **Education** | DSP | 82.81 ± 2.61 | 85.20 ± 0.64 | 71.70 ± 0.95 | Not Supported | 90.75 ± 0.01 | 81.50 ± 2.26 | 83.08 ± 1.10 | 84.90 ± 7.59 |
| | Geo | 72.96 ± 1.59 | 65.12 ± 4.07 | 57.90 ± 2.69 | | 71.92 ± 0.01 | 70.90 ± 2.45 | 80.44 ± 3.19 | 81.58 ± 0.57 |
| | HWTS | 73.93 ± 3.76 | 73.11 ± 2.13 | 68.63 ± 3.78 | | 82.89 ± 0.04 | 75.10 ± 11.67 | 72.59 ± 2.84 | 78.34 ± 0.95 |
| | VA | 74.90 ± 5.28 | 71.39 ± 3.38 | 74.37 ± 1.11 | | 77.80 ± 0.01 | 69.99 ± 8.83 | 71.43 ± 1.11 | 72.08 ± 3.71 |
| **Health** | B. Cancer | 89.70 ± 1.05 | 88.77 ± 7.31 | 80.52 ± 6.21 | Not Supp. | 78.19 ± 3.54 | 85.26 ± 1.03 | 84.66 ± 3.02 | 94.85 ± 1.25 |
| **Text** | AG News | 89.93 ± 3.32 | Not Supported | Not Supported | 88.73 ± 0.69 | 85.72 ± 5.31 | Not Supported | 87.25 ± 2.48 | 90.35 ± 1.07 |
| | SST | 91.12 ± 2.03 | Not Supported | Not Supported | 82.05 ± 0.56 | 88.21 ± 3.41 | Not Supported | 92.98 ± 0.88 | 91.75 ± 1.86 |
| **Synthetic** | OpenXAI | 86.67 ± 0.31 | 87.85 ± 1.31 | 83.67 ± 1.86 | Not Supp. | 89.51 ± 0.51 | 84.67 ± 4.04 | 90.83 ± 1.93 | 89.47 ± 2.89 |

Table 2: `InterpretCC` **Performance** (avg ± std) on EDU (balanced accuracy), Text, Health, and Synthetic (accuracy) datasets compared to a non-interpretable baseline and four intrinsically interpretable baselines. All 95% CI overlap with the non-interpretable base module for values in black. Colored values indicate significantly higher (green) or lower (red) performance than the base module. The reported Group Routing results are the best performing `ICC` variations from Table 3.

our sparsity criteria, we ensure that few groups are used to compute the output, making the Group Routing intrinsically interpretable at the group level, regardless of the types of models used as experts. Group Routing enables efficient inference without reducing the number of parameters available during training. During the training phase, we employ soft masking, allowing all weights $G(x)_i$ to remain non-zero, thus granting the model access to every expert. This approach allows the model to leverage the full set of parameters during training, enhancing the training efficiency. At inference time we switch to using a hard mask, making the weights sparse and allowing for interpretability.

# 4 EXPERIMENTAL SETTINGS

We apply `InterpretCC` to five domains: education, news classification, sentiment classification, healthcare, and synthetic data covering *Time Series*, *Text*, and *Tabular* inputs; all for classification tasks. For *Tabular* features, the input is a vector $x \in \mathbb{R}^n$. The mask in the Feature Gating is a sparse vector indicating which tabular feature to use and how important they are (if the weight is non-0) and the groups form a partition over the features. For *Text* features, the input is a sequence of $N$ tokens: $x = [t_1, t_2, \cdots, t_N]$. The mask is a sparse vector that indicates which token to use and each group consists of a subset of the tokens. Finally, we consider *Time Series* of $n$ features across $T$ timesteps: $x \in \mathbb{R}^{n \times T}$. We apply the same mask across all time steps for `InterpretCC` FG and GR.

**EDU** (*time series, education domain*). We predict student success in the early weeks of four massive open online courses (MOOCs), using students' clickstream data (see Table 5 in Appendix B for details about the courses). The raw clickstream input is transformed into weekly time-series features that have proven useful for student success prediction in previous literature (*e.g.* total video clicks, forum interactions). We select 45 input features used in multiple studies (Lallé and Conati, 2020; Boroujeni et al., 2016; Chen and Cui, 2020; Marras et al., 2021). For early prediction, we only use the first $40\%$ of time steps as input.

*Grouping:* To derive human-interpretable concepts from these features, we turn to learning science literature. In **routing by paper**, we create 10 distinct feature subsets based on handcrafted initial input features from 10 papers, directing each to a specific expert subnetwork. For **routing by pattern**, we organize features according to five learning dimensions identified by (Asadi et al., 2023; Mejia et al., 2022): effort, consistency, regularity, proactivity, control, and assessment-based (see Table 6 for a detailed feature classification). Thirdly, **routing by Large Language Model (LLM)**, uses GPT-4's capabilities to aid humans in feature grouping (Achiam et al., 2023). GPT-4 is prompted as an 'expert learning scientist' to group the features into self-regulated behavior categories that are easy to understand, which are then used to separate the features for `InterpretCC`. More details are included in Appendix C.1.

**AG News and SST** (*text, news and review domains*). For news categorization (**AG News**), we classify news into four categories ('World', 'Sports', 'Business', 'Sci/Tech') given a title and description of a real-world article (Zhang et al., 2015). We use 36,000 training samples and 3,000 test samples evenly distributed across categories. For sentiment prediction (Stanford Sentiment Treebank, **SST**), we use 11,855 sentences from movie reviews labeled by three annotators (Socher et al., 2013) and predict a binary sentiment from a sentence fragment.

*Grouping:* The `ICC` routing model assigns words to subnetworks using the Dewey Decimal Code (DDC) hierarchy of topics for book classification to create 10 subnetworks (see Table 7, Appendix

C.2 for more details) (Satija, 2013). Each word is encoded using SentenceBERT (Reimers and Gurevych, 2019) and assigned to a subcategory (*i.e.,* the word 'school' is assigned to the subcategory 'education' under category 300 for 'social sciences') and routed to the appropriate parent network.

**Breast Cancer** (*tabular, healthcare domain*). The Wisconsin Breast Cancer dataset identifies cancerous tissue from fine needle aspirate (FNA) images, with 30 features (10 per cell nucleus) and diagnoses (Malignant: 1, Benign: 0) for 569 patients (Wolberg et al., 1995).

*Grouping:* For the grouping logic, we group each cell nucleus in a separate subnetwork, enabling features representing the same part of the tissue sample to be considered together.

**Synthetic Dataset** (*tabular*) We use OpenXAI's synthetic dataset (Agarwal et al., 2022), which includes ground truth labels and explanations, indicating the subset of features influencing each label. This dataset comprises of 5000 samples, 20 continuous features, and two classes. It was created using the *SynthGauss* mechanism from five cluster neighborhoods (1000 points for each cluster), ensuring three desirable properties for assessing explanations: (1) feature independence, (2) unambiguous, well-separated local neighborhoods, and (3) an explanation for each instance.

*Grouping:* We group the feature space by assigning each feature to a cluster neighborhood based solely on the distribution of the training data. The average absolute value of each feature for each cluster is calculated, and the highest feature-cluster value determines the assignment.

## 5    EXPERIMENTAL RESULTS

Through the following three experiments, we demonstrate that our `InterpretCC` models do not compromise *performance* compared to black-box models and provide explanations that are *faithful* as well as *human-centered*. `InterpretCC` is designed for data that has meaning for humans (*i.e.,* interpretable features or meaningful raw data like text or lab measurements); however, we demonstrate it is also performant on a synthetic dataset with no interpretable features.

**Experimental Setup.** We run hyperparameter tuning and three different random seeds for each reported model (reproducibility details in Appendix F). Since **EDU MOOC** courses have a low passing rate (below 30%), and thus the dataset has a heavy class imbalance, we use balanced accuracy for evaluation. The other datasets are more balanced (**AG News**, **SST**, **Breast Cancer**, **Synthetic**), hence we use accuracy as our evaluation metric. We perform an 80-10-10 train-validation-test data split stratified on the output label, to conserve the class imbalance in each subset. In addition to `InterpretCC` Feature Gating and Group Routing, we also employ a `InterpretCC` Top-K expert network solution with k=2 for group routing. This approach is similar to existing mixture-of-expert approaches (Jiang et al., 2024; Li et al., 2022), except that their models make a layer-wise expert choice, which significantly reduces interpretability, while we make a global expert choice.

**Base Prediction Module.** We choose simple yet performant predictive modules reported in previous literature to isolate the difference in performance due to the interpretable architectures. For the EDU data, previous works use BiLSTMs on student behavioral data for best predictive performance (Swamy et al., 2022b; Marras et al., 2021). Thus, for comparative benchmarking, the most performant BiLSTM setting is used as a baseline model (Swamy et al., 2022a). For the AG News and SST datasets, we use fine-tuned `DistilBERT`[2] variations as baselines, also used in related works (Yang et al., 2019; HF Canonical Model Maintainers, 2022). For the Breast Cancer dataset, we use a fully connected network as reported by Agarap (2018).

**Interpretable Baselines.**   In addition to non-interpretable baselines, we compare `ICC` to three intrinsically interpretable methods: SENN (Alvarez Melis and Jaakkola, 2018), NAM (Agarwal et al., 2021), and FRESH (Jain et al., 2020). SENN generates concept-based explanations and is designed to learn *"interpretable basis concepts"* in parallel with the model optimization. Explanations consist of the concepts most similar to the input. NAM, a General Additive Model (Hastie, 2017), uses individual neural networks for each input feature to calculate feature weights. The model's output is the sum of these weights, and explanations are given by displaying the feature weights. The SENN Features architecture has three models working together, NAM has one subnetwork for each feature, and `ICC FG` has one model with two parts trained end-to-end. At inference, SENN and NAM also assign a score to each feature, which are then aggregated; `ICC` leverages cross-feature interactions with one score per model. FRESH is an extractive rationale architecture with three models: *supp*

---

[2]https://huggingface.co/distilbert/distilbert-base-uncased

| Dataset | Baseline | InterpretCC Group Routing | | |
|---|---|---|---|---|
| | | *Paper* | *Pattern* | *GPT-4* |
| **DSP** | 82.81 ± 2.61 | 82.37 ± 6.27 | 82.29 ± 3.72 | **84.90 ± 7.59** |
| **Geo** | 72.96 ± 1.59 | 69.64 ± 1.23 | **81.58 ± 0.57** | 81.19 ± 1.53 |
| **HWTS** | 73.93 ± 3.76 | **78.34 ± 0.95** | 72.34 ± 2.77 | 75.12 ± 4.17 |
| **VA** | 74.90 ± 5.28 | 69.88 ± 2.93 | **72.08 ± 3.71** | 70.98 ± 2.77 |
| **Average** | 76.65 ±3.31 | 75.56 ±2.85 | 77.57 ±2.69 | **78.05 ±4.01** |

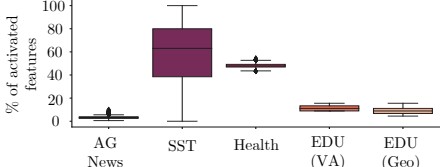

Table 3: **InterpretCC Group Routing Performance**: balanced accuracy (average ± std) on routing strategies (paper, pattern, GPT-4) for the EDU datasets in comparison to the non-interpretable baseline.

Figure 2: **InterpretCC Feature Gating Sparsity**: % of features activated per data point across five representative datasets.

generates importance scores, *ext* selects a contiguous text mask, and *pred* applies this mask on the inputs to make predictions. Designed for text data, FRESH's contiguous explanations are unsuitable for modalities like tabular or time-series data, where feature proximity lacks relevance.

## 5.1 EXP. 1: INTERPRETCC DOES NOT COMPROMISE ON PERFORMANCE

Table 2 shows the models' metrics (average accuracy, standard deviations, and 95% confidence intervals) across three iterations of model training for all eight datasets for InterpretCC Feature Gating, Group Routing, and Top-K Routing, as well as baselines of NAM, SENN Features, SENN Concepts, and a non-interpretable model (reflecting post-hoc explainer performance). We do not report results for text features with NAM and SENN, since the architecture change required to do so would no longer respect the original model design[3]; instead we report text results for FRESH. Additional sensitivity and architecture experiments can be found in Appendix D. These experiments show that while the performance of InterpretCC has overlapping 95% CIs while changing parameters, certain parameter settings have higher variability than others. For both education and health tasks, a $\tau$ of 10 and a Gumbel-Softmax threshold of around 0.7 to 0.8 are performant, sparse in activated features, and relatively stable.

InterpretCC Feature Gating statistically significantly improves performance with respect to the non-interpretable base prediction module for two EDU courses (DSP, HWTS) and the synthetic dataset. It shows comparable performance on all other datasets (indicated by the overlapping CIs) except the Breast Cancer dataset, suggesting that a higher number of the available features is necessary for performant prediction. The intrinsically interpretable baselines SENN, NAM, and FRESH never outperform the base model, and always under-perform or perform comparably to InterpretCC.

InterpretCC Group Routing outperforms (for the Geo course and Breast Cancer dataset) or performs comparably (95% CI overlap) to the fine-tuned, non-interpretable base module. It also consistently performs at least comparatively to SENN which, besides never outperforming the base model, has a relatively high variance. We further observe that the selected grouping method impacts performance (Table 3). We achieve a 10% increase in performance compared to the base model when grouping using patterns or GPT-4 for the Geo course. On average, over the four EDU courses, the automated LLM grouping and the pattern-based human-defined grouping perform comparably, showing that using automated grouping methods does not mean compromising on performance.

> InterpretCC performs comparably to black-box models and outperforms intrinsically interpretable baselines across diverse benchmarks.

## 5.2 EXP. 2: INTERPRETCC PROVIDES FAITHFUL AND USER-FRIENDLY EXPLANATIONS

Table 4 showcases the faithfulness of InterpretCC models in comparison to intrinsically interpretable models SENN and NAM, as well as three post-hoc explainers on top of a non-interpretable model (Integrated Gradients (IG) (Sundararajan et al., 2017), LIME (Ribeiro et al., 2016), and SHAP (Lundberg and Lee, 2017)). We examine the relationship of the explanations to underlying data patterns from the synthetic dataset as well as to the ground truth of the underlying model (see Appendix I.2 for detailed descriptions of the metrics). All 95% CI overlap in Ground Truth Alignment

---

[3]For NAM, one network (LLM) would be required per word, as the words are distinct for each instance; it does not support text grouping. For SENN, LMs would need to embed each word, then simultaneously be trained to represent concepts and pick representative words, with new metrics for choosing prototypical examples.

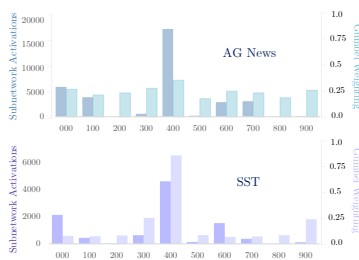

Figure 3: **AG News** and **SST**: # of ICC subnetwork activations (left) vs. avg. activation weights (right), grouped by subnetworks based on the Dewey Decimal Code.

| Model | Relationship to Underlying Data Patterns | | | Relationship to Model |
|---|---|---|---|---|
| | *Ground Truth Alignment (GTA)* | *Rank Agreement (RA)* | *Feature Agreement (FA)* | *Ground Truth Faithfulness (GTF)* |
| ICC FG | **94.84 ± 4.11** | **60.00 ± 12.65** | 87.99 ± 16.00 | always **100** |
| ICC GR | 89.51 ± 0.56 | 44.00 ± 14.97 | 76.00 ± 23.32 | always **100** |
| SENN Features | 85.83 ± 2.56 | 27.99 ± 9.80 | 88.01 ± 9.79 | always **100** |
| SENN Concepts | 65.19 ± 9.23 | 52.00 ± 20.39 | 80.00 ± 21.90 | always **100** |
| NAM | 87.39 ± 2.45 | 40.00 ± 17.89 | 76.00 ± 14.97 | always **100** |
| IG | 87.28 ± 1.72 | 56.00 ± 14.97 | **91.99 ± 16.00** | ≤ 100 |
| LIME | 84.75 ± 1.83 | 44.00 ± 23.32 | 64.00 ± 14.97 | ≤ 100 |
| SHAP | 83.47 ± 1.03 | 36.00 ± 8.00 | 52.00 ± 20.39 | ≤ 100 |

Table 4: **Synthetic faithfulness evaluation** across eight models on explanations using metrics presented in OpenXAI (Agarwal et al., 2022). GTA, RA, and FA (K=5) refer to the explanations' relationship to underlying patterns in the synthetic data. GTF refers to bounds of the explanations' relationship to the model's decision process. All 95% CIs overlap between the best performing intrinsically interpretable and non-interpretable model.

(how close the explanation is to the underlying synthetic data feature patterns, similarly to (Liu et al., 2021)), Rank Agreement (RA) and Feature Agreement (FA) from Agarwal et al. (2022), except for SENN Concepts performing statistically significantly worse than `InterpretCC` Feature Gating in GTA. This indicates that `ICC`'s identification of important features, and of their order of importance, is close to the underlying data patterns and is either on par or better than both interpretable and non-interpretable models. In terms of predictive performance on the synthetic dataset (last row of Table 2), `ICC` holds the top three best performing models (89.47, 89.51, 90.83) and the `ICC` FG variation is significantly more performant than the non-interpretable base model, demonstrating the models' ability to capture signal on this dataset.

Our models provide concise and hence user-friendly explanations through sparse feature (group) activations (Miller, 2019). Figure 2 shows the percentage of activated features for `ICC` Feature Gating. For **EDU**, only about 10% of the 45 features are activated with low variance. For **Breast Cancer**, 39.7% of the features are activated per data point. Unlike other datasets, text-based datasets have a variable number of features (words). In **AG News**, with an average of 35 words per article, only a small percentage is activated, while in **SST**, which has shorter sentences (7 words on average), 59.8% of features are selected with high variance. This achieved sparsity, especially in **EDU** and **AG News**, highlights the most important features. This contrasts with post-hoc explainers, which tend to select a broader range of features (*e.g.,* LIME and SHAP for EDU (Swamy et al., 2022b)).

`ICC` Group Routing activates different subnetworks with different weights for each data point. Figure 3 illustrates the number of activations and the average weight for each subnetwork for the text data sets (see Appendix H for detailed analysis on the Breast Cancer and EDU datasets). For **AG News** (Figure 3 top), the average activation weight is similar across all subnetworks (min 0.10, max 0.21). However, some subnetworks are activated much more frequently (400 - Language: 18, 335 times). This indicates that most data points will be routed through the same subset of subnetworks, while the remaining subnetworks are important for specific data points only. **SST** (Figure 3 bottom) shows similar subnetwork activation patterns. However, in contrast to **AG News**, the distribution of average weights is not uniform: only three networks are activated with weights larger than 0.15. We suspect the high weights for subnetwork 400 in Figure 3 reflect words that DDC has little relation to in the SentenceBERT embedding space.

Examples of EDU domain explanations are in Appendix E, Figures 9 and 10, with additional AG News examples in Figure 15. We also compare `ICC` behavior across three grouping strategies for the DSP course (Appendix H.3.2) and analyze network sparsity at different prediction horizons (H.3.1). Lastly, we show the variation in feature group selection across MOOCs, highlighting `ICC`'s adaptability regardless of grouping method (Appendix H.3.3).

> `InterpretCC` provides sparse and hence user-friendly explanations, while not compromising on explanation faithfulness.

## 5.3 EXP. 3: INTERPRETCC EXPLANATIONS ARE PREFERRED BY HUMANS

**Setting.** To validate the user-centeredness of our approach, we conducted a user study comparing `InterpretCC`'s explanations with the ones from other intrinsically interpretable methods, SENN

and NAM. We focused on the education domain and time series input type, using the DSP course in the EDU dataset. We trained `InterpretCC` Feature Gating and Group Routing (with pattern-based feature grouping strategy, since it is heavily grounded on expert knowledge) as well as SENN and NAM and randomly selected four test samples (*i.e.,* four students) for prediction. In designing the study, we conducted four sequential pilots with eight learning scientists who were unaware of which method we presented in this paper. All explanations were simplified for a non-technical audience and followed consistent templates.

We recruited 56 teachers using `Prolific`, (see Appendix E for detailed information about the participants' demographics and backgrounds (Figure 8) and the content of the study). We showed them each model's prediction of the student's success or failure along with its explanation. The explanations were given as a short text and a graph showing the features and concepts used by the model. Note that the choices we made for the presentation of the explanations might have an influence on the participants' perception of the explanations. Examples of how the explanations were presented and a discussion on this limitation can be found in Appendix E, Figures 9, 10, 12 and 11. With our study design, we aimed to highlight each framework's strengths without excessive post-processing. For instance, keeping only the top five features of NAM would be unfair, as other features also contribute to its predictions. Instead, we emphasized the top five features that each contribute positively, negatively, or not at all to the prediction, highlighting its advantage (distinct insights into feature impact) over `InterpretCC`, while ensuring sparsity. We significantly post-processed the explanations of SENN and NAM to provide them in a format understandable for a non-technical audience and iterated on the visualizations using a human-centered design process (Cooley, 2000). We asked participants to compare these explanations according to five criteria (aligned with (Frej et al., 2024)), and to rank the criteria in terms of importance: **Usefulness**: This explanation is useful to understand the prediction. **Trustworthiness**: This explanation lets me judge if I should trust the model. **Actionability**: This explanation helps me know how to give feedback to the student.**Completeness**: This explanation has sufficient detail to understand why the prediction was made. **Conciseness**: Every detail of this explanation is necessary.

|  | NAM | SENN | ICC GR | ICC FG | Weight |
|---|---|---|---|---|---|
| Usefulness | 3.25 ±0.98 | 3.3 ±1.11 | 3.53 ±1.11 | **3.88 ±0.94** | 0.28 |
| Trustworthiness | 3.28 ±0.93 | 3.64 ±0.92 | 3.36 ±1.06 | **3.78 ±0.9** | 0.23 |
| Actionability | 3.08 ±0.96 | 3.25 ±1.06 | 3.37 ±1.04 | **3.77 ±0.95** | 0.21 |
| Completeness | 3.18 ±1.02 | **3.76 ±1.09** | 3.1 ±1.19 | 3.67 ±1.07 | 0.16 |
| Conciseness | 3.13 ±1.06 | 2.82 ±1.31 | **3.72 ±1.06** | 3.68 ±1.05 | 0.12 |
| Global | 3.2 ±0.81 | 3.38 ±0.85 | 3.41 ±0.88 | **3.78 ±0.77** |  |

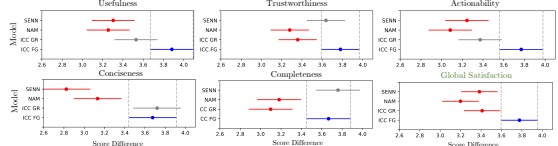

Figure 4: **Model score for each user study criterion** (average ± std) and criteria weight according to users' ranking. All scores range from 1 (lowest) to 5 (highest).

Figure 5: **Tukey's Honest Significant Difference (HSD) Test** for each user study criterion and overall global satisfaction. ICC FG (with highest overall satisfaction in Table 4) is in blue.

**Results.** Using Friedman's Chi-Square test, we verify that the ranking of the criteria is consistent among the participants ($p < 0.05$). We report normalized criteria importance in Table 4 (last column). We observe that Usefulness, Trustworthiness and Actionability are consistently ranked the highest by the participants. We compute the *Global satisfaction* score as a weighted average of the criteria as a global satisfaction measure for each model (Table 4, last row). We additionally conduct pairwise Tukey's Honest Significant Difference (HSD) tests (Fig. 5) to determine the statistical significance of the differences between the means of user preferences per criteria.

We observe that `ICC FG` ranks the highest in the top 3 most important criteria as well as in global satisfaction. Furthermore, `ICC GR` ranks second in Usefulness and Actionability and first in Conciseness. Overall, `InterpretCC` models are favored over interpretable baselines in 4 out of 5 criteria and in terms of global satisfaction. An ANOVA performed separately for each criterion as well as the global satisfaction measure (Table 9, Appendix E.3), indicates that there is indeed a significant difference between the models for each criterion. Tukey's HSD tests confirm that `ICC FG` significantly outperforms NAM and SENN on all criteria but completeness (Fig. 5).

Participants prefer `InterpretCC` explanations in terms of usefulness, trustworthiness, actionability, and conciseness over other intrinsically interpretable models.

## 6    DISCUSSION AND CONCLUSION

We proposed `InterpretCC`, a family of intrinsically interpretable models that puts human understanding at the forefront of the design. Through our experiments on feature gating and group routing (mixture-of-expert) models, we demonstrated that our modular architectures optimize for interpretability but do not compromise on performance. In a real-world setting, we showed that `InterpretCC` models are preferred over other intrinsically interpretable models in 4 of 5 explainability criteria.

`InterpretCC` is a locally intrinsic explanation framework that creates explanations that are specific to the input point and guaranteed to be faithful to the model. In our architecture, the discriminator network is a black-box model. This is by design, to minimize explanation complexity. We believe the hierarchical prediction logic enabled by `InterpretCC` is the type of explanation that a user wants: "Which concepts/features were used to make the prediction, and how important are they to the prediction (weighted sum)?" If the discriminator network was a glass-box model, it would answer the question: "why were these concepts/features selected for the prediction?". The first explanation is directly actionable, while the second type of explanation is not. We therefore do not seek to answer this second question with our architecture. However, it would be possible for `InterpretCC` to have an interpretable discriminator network, as discussed further in Appendix I.3.

`ICC FG` is best when individual features are important for actionable decisions based on the explanation. However, it can lose sparseness if all features are equally important: consider the extreme case where the prediction is a sum over the full feature space and all features are independent. `ICC GR` requires more human effort than `ICC FG` and is, therefore, more suitable for scenarios with cross-feature dependencies and where broader concepts are more actionable than individual features.

`ICC GR` model's global mixture-of-experts design specializes subnetworks on subsets of features allows them to learn granular patterns (Appendix D.3). Combining these specialists enhances prediction compared to a monolithic network that might miss underlying patterns (Table 2). Expert-informed feature groupings help the model avoid overfitting to correlations that do not generalize at inference time. For `ICC FG`, adaptive sparsity (a few features per instance) improves prediction quality by reducing noise and optimizing the interpretability-accuracy tradeoff.

User-defined feature groups aim at deriving explanations useful to the user, but might compromise performance if the user specifies a grouping that carries minimal signal. Regardless of the grouping, `InterpretCC` optimizes for explanation actionability and understandability over performance. We believe that an accurate prediction at the cost of explanation usefulness is not worthwhile in an applied setting. It is possible that `ICC` explanations could be misleading, as concepts used in the explanation could lead to a correlation that was not intended (Zheng et al., 2021; Jacovi and Goldberg, 2021). In these cases, we view `ICC` explanations through the lens of auditing model behavior (Yadav et al., 2022), and encourage human intervention. `InterpretCC`'s user-centric advantages are highlighted when the input space is human-interpretable. However, for domains that are hard to obtain expert knowledge, we envision `ICC` increasingly leveraging LLM-extracted features, reducing the necessity of human effort for human-centric explanations (Malberg et al., 2024; Baddour et al., 2024).

We acknowledge that the presentation of explanations in our user study (Section 5.3) has influence over our preliminary results on the user perception of `InterpretCC`. We conducted extensive iterations with eight pilot participants to mitigate study design bias. We note that any imbalance in wording is not necessarily in favor of our method; for instance, users found SENN's explanations more complete than `InterpretCC` (Fig. 5). The user study prioritizes the diversity of study participants and quality of responses over the number of samples evaluated; the task is mentally intensive and we found a longer study can cause a drop in participant attentiveness. An extensive study over many different tasks and domains of expertise is necessary for generalizable conclusions.

For our text experiments, we fine-tune twenty `DistilBERT` models as experts (ten for each task). For more complex tasks, for example requiring long context size, multi-step reasoning ability, or strong prior domain-specific knowledge, `DistilBERT` can be swapped with larger decoder models, either through fine-tuning or in-context learning. Parameter-efficient fine-tuning such as LoRA (Hu et al., 2021) would allow fine-tuning even large LMs with limited computational cost, while in-context learning would use the same model instance for each feature and feature group.

Overall, we encourage the machine learning community to design models for interpretability at many different granularities and user-specified requirements. `InterpretCC` provides one such family of models as a tradeoff between human specification, explanation certainty, and performance.

# 7 ACKNOWLEDGEMENTS

We kindly thank the Swiss State Secretariat for Education, Research and Innovation (SERI) for supporting this project, and Dr. Tanya Nazaretsky, Bahar Radhmehr, and Paola Mejia Domenzain for helpful discussions. This project has received funding from the European Union's Horizon 2020 Research and Innovation programme under grant agreement No. 101017915 (DIGIPREDICT).

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
