## IMPACT STATEMENT

This paper introduces work aimed at advancing the field of human-centered machine learning, addressing the critical balance between interpretability and model performance in neural networks. We demonstrate that these aspects do not necessarily need to be a trade-off, particularly in fields where AI can influence decisions that can have effects on people's lives, such as finance, healthcare, and education. Focusing on the applications presented in this paper, in healthcare, the ability to understand and trust AI-driven recommendations can greatly improve care delivery. In education, the application of interpretable models reduces the risks of unintended biases, help users to identify actionable interventions, and increases trust.

A key impact of interpretability is regulatory compliance, where guidelines and regulations around AI usage are increasing (e.g. through the EU AI Act). Models like `InterpretCC` can help industries and organizations meet these legal standards by increasing the transparency of their AI-based decision-making tools.

`ICC` enhances trust and human-centered actionability in predictive tools by clearly listing the features impacting predictions. This empowers decision-makers to take informed actions. Different contexts and applications have different interpretability needs, such as different levels of granularity, or different sets or actionable concepts. `ICC` Group Routing allows users to define their own set of features, adapted to their needs. Moreover, by identifying a finite set of key features, `ICC` may reduce spurious predictions and allow users to better understand the causes of unexpected model behavior. Overall, an interpretable model like `ICC` allows users to assess the model's robustness more easily and faithfully on the scale of a individual's decision.

Our tool also offers the potential to identify and mitigate biased behaviors within models, particularly if discriminatory predictions arise from certain feature uses. As different contexts may require varying levels of interpretability, the Group Routing feature of `ICC` allows users to define and adjust feature sets according to their specific needs, ensuring flexibility and relevance in diverse applications.

Comparable in societal and technological impact to other interpretable-by-design methods, such as those discussed by Agarwal et al. (2021), `ICC` represents a significant step forward in making AI more accessible and understandable for users and practitioners alike.

We acknowledge that despite the versatility of `ICC` across various data types, domains, and datasets demonstrated in this work, we cannot guarantee that it will always perform on par with or better than baseline models. It is important to note that the models we enable are not safe for direct use in production tasks, such as detecting breast cancer, without further validation and adaptation by medical professionals. We encourage researchers and practitioners to adopt explainable AI methods.

## A TAXONOMY OF EXPLANATION DESIGN CRITERIA

We include a detailed taxonomy of terms used in Table 1, inspired by Speith (2022); Schwalbe and Finzel (2024); Swamy et al. (2023b); Pinto and Paquette (2024).

**Granularity:** The granularity (small details or broad concepts) of the explanation Miller (2019); Swamy et al. (2023b); Pinto and Paquette (2024).

- **Feature:** Explanations are made using the input features.
- **Concept:** Explanations are made using concepts (a higher level grouping over the input space).

**Basis:** The type of input used by the XAI method/model to create explanations (Saeed and Omlin, 2023; Ghorbani et al., 2019b).

- **Use all input features:** The explanation uses all of the raw inputs to the model.
- **Concepts:** Either the user defines a concept by specifying examples, the model automatically selects concepts from the feature space, or there is some external definition of concepts grouped over the feature space (user or LLM-specified).

**Stage:** The extent to which a model incorporates the explanation in the predictive process (Ali et al., 2023; Speith, 2022; Schwalbe and Finzel, 2024).

- **Explain then predict:** The explanation is directly used as part of the prediction process, influencing the model's output.
- **Explain from model internals:** The explanation reflects the weights or internal mechanisms of the model but is not directly used for prediction (gray box).
- **Explanation not used in model:** The explanation is completely separate from the model's internals and has no influence on the predictive process (black box).

**Sparsity:** The conciseness of the explanation relative to the amount of the input space (features, concepts) used in the prediction (Sun et al., 2024; Ayoobi et al., 2023).

- **Sparse:** The explanation uses a minimal amount of features/concepts for prediction.
- **Not Sparse:** The explanation uses more than a minimal amount of the features/concepts for prediction.

**Faithfulness:** Alignment between the explanation and the model behavior, also known as fidelity (Lyu et al., 2024; Dasgupta et al., 2022).

- **Guaranteed:** The explanation aligns to the model behavior with certainty.
- **Aligned with concepts:** The explanation aligns with the concepts used but is not guaranteed to align with the model behavior.
- **Approximation:** The explanation is not guaranteed to reflect the model behavior.

## B  ADDITIONAL DETAILS ON DATASETS

Here, we provide additional statistics regarding each dataset used in our study. In particular, we highlight their availability details and terms of use.

**EDU.** We predict student success during the early weeks of four massive open online courses (MOOCs), using students' clickstream data (see Table 5 for details about the courses). Contrary to the other 4 datasets, this dataset is kept private for student privacy reasons, as required by HREC 058-2020/10.09.2020 and HREC 096-2020/09.04.2022.

| Title | Identifier | Topic | Level | Language | No. Weeks | No. Students | Passing Rate [%] |
|---|---|---|---|---|---|---|---|
| Digital Signal Processing | DSP | CS | MSc | English | 10 | 4,012 | 23.1 |
| Éléments de Géomatique | Geo | Math | BSc | French | 11 | 452 | 45.1 |
| Household Water Treatment and Storage | HWTS | NS | BSc | French | 5 | 2,438 | 47.2 |
| Villes Africaines | VA | SS | BSc | En/Fr | 12 | 5,643 | 9.9 |

Table 5: Course Details and Statistics.
Topic abbreviations: Math: Mathematics; NS: Natural Science; CS: Computer Science; SS: Social Science; Arch: Architecture; Bus: Economics and Business.

**AG News** is a news classification dataset, where given a title and description of a real-world article, it has to be classified into one of the four categories: 'World', 'Sports', 'Business', 'Sci/Tech' (Zhang et al., 2015). It is freely available at the following location: `http://groups.di.unipi.it/~gulli/AG_corpus_of_news_articles.html`, but only for non-commercial use.

**SST.** The Stanford Sentiment Treebank dataset aims at predicting the sentiment from a sentence fragment sourced from a movie review. The dataset is freely available here: `https://huggingface.co/datasets/sst`. This popular benchmark is an extension of the Movie Review Database (MRD) (Socher et al., 2013). It includes two sets of labels: one for binary sentiment classification and one for multiclass. We use binary classification to demonstrate a different setting than the multiclass classification of AG News.

**Breast Cancer.** The Wisconsin Breast Cancer dataset attempts to identify the presence of cancerous tissue from an image of a fine needle aspirate (FNA) of a breast mass (Wolberg et al., 1995). This dataset is freely available here: `https://archive.ics.uci.edu/dataset/17/breast+cancer+wisconsin+diagnostic`, and is licensed under a Creative Commons Attribution 4.0 International (CC BY 4.0) license.

**Synthetic Dataset.** We generate 5000 points of OpenXAI's synthetic dataset (Agarwal et al., 2022) through the class `generate_gaussians` with 5 cluster centers, which includes both ground truth labels and ground truth explanations, and is available here: `https://github.com/AI4LIFE-GROUP/OpenXAI` (under MIT license).

## C  INTERPRETCC GROUP ROUTING SCHEMA

In the following, we discuss the exact grouping schematic employed on the 4 EDU datasets (including three schemes) and the 2 text datasets.

**Group Specification Guidance**: In `InterpretCC`, we design towards the user's actionability of the resulting explanation. Therefore, if the knowledge that a feature is important can lead to a specific action, and if this action is the same one that should be taken for other features, then those features should be grouped together. From the modeling perspective, grouping features together in a concept means that their shared predictive potential should be leveraged, and likely this is more important for one type of features than another. We maintain that a feature should not be placed in multiple groups to have the faithfulness guarantees that are a strength of `InterpretCC`. However, in a rare and specific case where two actions must be taken based on the feature, or it is too difficult to decide which concept the feature belongs in, it is always possible to 1) put that feature in its own subnetwork (and have it be selected alongside any of the other feature groups), or 2) combine the two subnetworks into a larger concept.

Concretely, for `ICC` GR, users specify a grouping that separates individual features into concepts (which are groups of features). User selected features and user definitions of features are not required in `ICC` FG. For concept specification, users describe feature groupings in the form of a list or dictionary with which features are assigned to which network. In training, the discriminator takes in all the features as input and decides which of the C concepts to activate. In prediction, there are C predictive modules (one for each concept), each with only the features that are mapped to that concept as input. If a specific concept is activated, then only the features assigned to that concept (by a user) are sent to that subnetwork, and this is used for the eventual prediction.

### C.1  FEATURE GROUPING SCHEMA FOR EDU DATASETS

The data is composed of student clickstream data from four different MOOCs. The raw clickstream input is transformed into 45 weekly time-series features that have proven useful for student success prediction in previous literature (e.g. total video clicks, forum interactions).

**Routing by pattern.** We organize features according to four learning dimensions identified by Asadi et al. (2023) and detailed in Table 6—effort, consistency, regularity, proactivity, control, and assessment—based on Mejia et al. (2022), with a focus on these dimensions in an extended experiment.

**Routing by Paper.** In routing by paper, we create 4 distinct feature subsets based on handcrafted initial input features from 4 learning science research papers highlighted as predictive for MOOCs in Swamy et al. (2022a). We filter each feature set to a expert subnetwork for the research paper the set originated from Lallé and Conati (2020); Marras et al. (2021); Chen and Cui (2020); Boroujeni et al. (2016).

**Routing by GPT-4.** To leverage LLMs for automated feature groupings, we use the following prompt alongside the feature descriptions from the appendix table of Swamy et al. (2022a).

**Prompt**:

> You are a talented learning scientist, specializing in self-regulated learning behavior. You want to group the following features together into smaller groups of logical concepts based on educational theory with headings. What are your feature groupings?
>
> **Response**:
>
> Certainly, based on educational theory and self-regulated learning behavior, I would group the provided features into the following logical categories:

- Temporal Engagement:
  - DelayLecture
  - RegPeakTimeDayHour
  - RegPeriodicityDayHour
- Study Patterns:
  - NumberOfSessions
  - AvgTimeSessions
  - TotalTimeSessions
  - StdTimeSessions
  - StdTimeBetweenSessions
- Video Engagement:
  - AvgWatchedWeeklyProp
  - StdWatchedWeeklyProp
  - AvgReplayedWeeklyProp
  - StdReplayedWeeklyProp
  - AvgInterruptedWeeklyProp
  - StdInterruptedWeeklyProp
- Competency and Learning Progress:
  - CompetencyStrength
  - CompetencyAlignment
  - CompetencyAnticipation
  - ContentAlignment
  - ContentAnticipation

- Interaction Patterns:
  - TotalClicks
  - TotalClicksProblem
  - TotalClicksVideo
  - TotalClicksWeekday
  - TotalClicksWeekend
  - TotalTimeProblem
  - TotalTimeVideo
  - TotalClicksVideoLoad
  - TotalClicksVideo
  - FrequencyEventVideo
  - FrequencyEventLoad
  - FrequencyEventPlay
  - FrequencyEventPause
  - FrequencyEventStop
  - FrequencyEventSeekBackward
  - FrequencyEventSeekForward
  - FrequencyEventSpeedChange
  - AvgSeekLength
  - StdSeekLength
  - AvgPauseDuration
  - StdPauseDuration
  - AvgTimeSpeedingUp
  - StdTimeSpeedingUp
- Assessment Behavior:
  - StudentSpeed
  - StudentShape

| Dimensions | Corresponding measures | Student patterns |
|---|---|---|
| **Effort** | Total time online
Total video clicks | Higher intensity
Lower intensity |
| **Consistency** | Mean session duration
Relative time online
Relative video clicks | Uniform
First half
Second half |
| **Regularity** | Periodicity of week day
Periodicity of week hour
Periodicity of day hour | Higher peaks
Lower peaks |
| **Proactivity** | Content anticipation
Delay in lecture view | Anticipated
Delayed |
| **Control** | Fraction time spent (video)
Pause action frequency
Average change rate | Higher intensity
Lower intensity |
| **Assessment** | Competency strength
Student shape | Higher intensity
Lower intensity |

Table 6: **EDU Routing by Pattern** uses learning dimensions from Mejia et al. (2022) to create interpretable feature groupings.

| Code | Field of Study |
|---|---|
| **000** | Computer Science, Information and General Works |
| **100** | Philosophy and Psychology |
| **200** | Religion |
| **300** | Social Sciences |
| **400** | Language |
| **500** | Pure Science |
| **600** | Technology |
| **700** | Arts and recreation |
| **800** | Literature |
| **900** | History and geography |

Table 7: **Text Routing** by the Dewey Decimal Classification system Scott (1998). Each code represents a subnetwork in the text variations of the `InterpretCC` framework.

Each grouping represents a different aspect of self-regulated learning behavior, focusing on how students engage with learning resources, interact with content, demonstrate competency, and approach assessments. This categorization aligns with principles of self-regulated learning and can help in analyzing and understanding students' behaviors and strategies within an educational context.

## C.2 Feature Grouping Schema for Text Datasets

For news categorization (AG News) and sentiment prediction (SST) feature grouping, we assign words to subnetworks. For this, we use the Dewey Decimal Code (DDC) for librarians and its hierarchy of topics for book classification to create 10 subnetworks, as showcased by topic in Table 7 (Satija, 2013). Each word is assigned to a subcategory (i.e. the word 'school' is assigned to the subcategory 'education' under category 300 for 'social sciences') and routed to the appropriate parent network. The decision to use the DDC was to use subnetworks that were standardized, pervasive in daily life and clearly human-understandable. To conduct this assignment, we utilize SentenceBERT to encode the subtopics for each DDC heading (i.e. all of 010, 020, 030, etc. for the category 000) (Reimers and Gurevych, 2019). The choice of SentenceBERT is motivated towards capturing the broader context of multi-word category headings in a lightweight model. During training and inference, we again use SentenceBERT to encode each word in the input instance, then assign each word to the most similar DDC category in embedding space with cosine similarity.

## D   Hyperparameter Sensitivity and Architecture Validation Experiments

We examine the sparsity and Gumbel Softmax hyperparameters, and how they impact the `InterpretCC` model performance.

### D.1   Sparsity Criterion Experiments

For the feature gating architecture to further improve interpretability, we would like the network to learn sparse feature activations. That is, for a given input $x$ we would like to reduce the number of features that affect the model prediction. To achieve this we apply regularization to the generated feature mask.

One natural choice to enforce sparse feature activations is to apply $L_1$-norm regularization to the feature mask, which penalizes a high number of nonzero elements. Another choice is to use annealed regularization as presented by Verelst and Tuytelaars Verelst and Tuytelaars (2020), which might aid the model to first work through a prediction optimization phase that is not confounded by any additional error terms before moving towards a sparsity-enforcing phase.

We experiment with annealing L1 and L1 regularization across four courses, and find that traditional L1 regularization is more stable (at least in the time-series setting). The Baseline BiLSTM results are not reported as confidence intervals here as they are directly sourced from a recent benchmarking paper by Swamy et al., with confirmed similarity by Asadi et al. Swamy et al. (2022a); Asadi et al. (2023). We reproduce this benchmark above with similar values in Table 8.

| EDU Dataset | Baseline | InterpretCC Feature Gating | |
|---|---|---|---|
| *40% EP* | *BiLSTM* | *Annealing* | *L1* |
| DSP | 82 | 87.76 +/- 3.12 | 90.75 +/- 0.01 |
| Geo | 76.2 | 81.13 +/- 5.39 | 71.92 +/- 0.01 |
| HWTS | 72 | 77.58 +/- 0.01 | 82.89 +/- 0.04 |
| VA | 73.8 | 84.81 +/- 0.01 | 77.80 +/- 0.01 |

Table 8:  Annealing L1 regularization in comparison with L1 regularization across EDU datasets.

### D.2   Gumbel Softmax Hyperparameter Sensitivity

We explore the effects of varying the Tau and Threshold parameters during training for `InterpretCC` FG (top two plots) and GR (bottom two plots) using the DSP course dataset. This dataset was selected due to its extensive use as a benchmark for explainability in education research Swamy et al. (2023a). We examine the variability over a tuned discriminator layer (batch

size 8, and layer size 32), with additional configurations detailed in Appendix F. While BAC/ACC remains similar across settings, optimizing the Threshold in conjunction with Tau is crucial for performance; often times changing the threshold is quite a variable experience. Notice the changing impact on BAC across different thresholds as Tau changes.

The thresholds used were 0.5, 0.6, 0.65, 0.7, 0.725, 0.75, 0.775, and 0.8, while the tau values were 0.5, 1.0, 5.0, and 10.0.

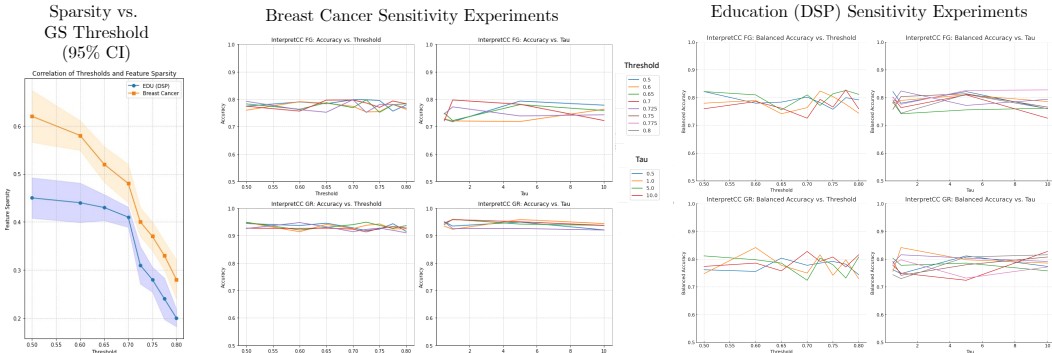

Figure 6: `InterpretCC` **Gumbel Softmax Hyperparameter Sensitivity Experiments.** We examine changing Tau and Threshold parameters during training for `InterpretCC` FG (top two plots) and GR (bottom two plots) for the DSP course dataset and the Breast cancer dataset.

All 95% CIs overlap for the experiments for Figure 6. They are omitted from the plots for readability. Significantly high variation (over 0.1) include the following settings:

- `InterpretCC` FG 0.5 threshold, 1.0 Tau with 0.183 standard deviation
- `InterpretCC` FG 0.6 threshold, 5.0 Tau with 0.122 standard deviation
- `InterpretCC` FG 0.65 threshold, 10.0 Tau with 0.151 standard deviation
- `InterpretCC` GR 0.5 threshold, 1.0 Tau with 0.107 standard deviation
- `InterpretCC` GR 0.6 threshold, 5.0 Tau with 0.114 standard deviation

GR architectures are on average 0.043 more stable (less variable) than FG architectures. These experiments show that while the performance of `InterpretCC` has overlapping 95% CIs while changing parameters, certain parameter settings have higher variability than others. For both education and health tasks, a $\tau$ of 10 and a Gumbel-Softmax threshold of around 0.7 to 0.8 are performant, sparse in activated features, and relatively stable. Notably, the results for the Breast Cancer dataset are less variable than the DSP dataset, which shows the sensitivity of parameters is domain dependent.

### D.3 ARCHITECTURE VALIDATION ANALYSIS

**"Do we genuinely achieve specialized networks?"** Jacobs (1997) demonstrates that mixtures-of-experts architectures can optimize the bias-variance trade-off by specializing subnetworks for specific regions of the input space. Similarly, Jiang and Tanner (1999) prove that the identifiability of mixtures-of-experts models depends on their parameterization. In the following experiment, we show that `InterpretCC` subnetworks specialize to information that are routed to them, and are worse at predicting on information that is not routed to them.

The experiment, conducted across three different use cases (SST: text, DSP: time series, Breast Cancer: tabular) of `InterpretCC`, demonstrate that subnetworks predict more strongly in on data that has been routed to them as opposed to data that is supposed to be routed to other subnetworks. Specifically, for DSP, the route-by-pattern networks predict 22.88% +/- 7.56% more accurately (balanced accuracy) on points sent to them. For SST, the subnetworks specialize even more strongly (23.59% +/- 7.86%). For Breast Cancer, the cell-based grouping predicts 11.88% +/- 5.57% better when routed to the same network. The heatmaps in Figure 7 show the prediction rates of subnetworks

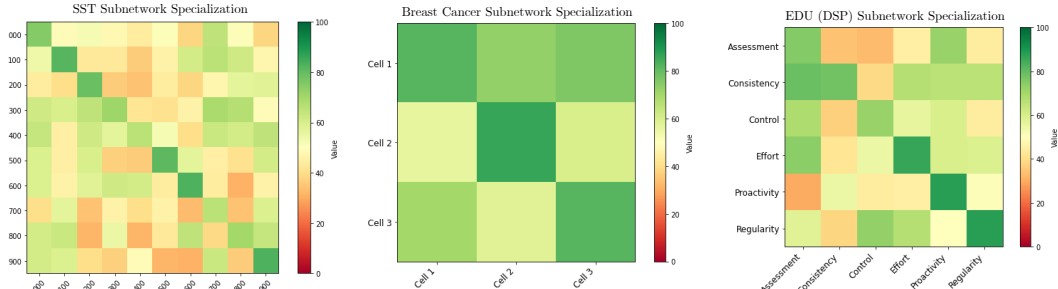

Figure 7: Validation of Specialized Subnetworks: SST, Breast Cancer, and DSP datasets.

(x-axis) on data routed from every other subnetwork (y-axis). Note that across all three use cases, the diagonal performance is the strongest.

**"Do these networks contribute to improved prediction accuracy?"** As proven by Jordan and Jacobs (1994), mixture-of-experts are a strongly predictive architecture, analogous to the case of ensemble models over monolithic models. The authors show that hierarchical mixtures-of-experts (HMoE) can recursively partition the input space and train effectively using the Expectation-Maximization (EM) algorithm. Empirically, in comparison to post-hoc explainability approaches on top of a non-interpretable base module, we show performance gains (Table 2) for `InterpretCC` GR in 7 out of 8 datasets, statistically significant higher performance in the Geo and Breast Cancer settings, and comparable performance (overlapping 95% CIs) in all other settings; this directly shows that using specialized subnetworks is superior to the non-interpretable baseline approach. As `InterpretCC` FG showcases simply the sparsity requirement with no subnetwork logic, and GR has higher performance than FG results in 4 settings, we know that the subnetworks can be helpful. Similarly, over the non-interpretable base module, FG statistically significantly beats performance in DSP, HWTS, and Synthetic data cases, showcasing the benefits of adaptive sparsity.

The comparison to NAM showcases the benefits of combining important features together in the predictive module instead of having a separate subnetwork for every feature. The comparison to SENN showcases the benefit of expert-specified routing logic as opposed to automated concept selection. The new comparison to FRESH showcases the benefit of selecting a concept-based explanation over a contiguous explanation. Hazimeh et al. (2021) proposes and proves that an architecture with similar foundations in conditional computation achieves efficient and differentiable sparse subnetwork selection, improving task performance and computational efficiency in multi-task learning. They do not focus on an interpretability objective.

**"Do these networks contribute to improved interpretability?** Indeed, intrinsic explanations (through hierarchical models or gating) do contribute to increased interpretability (Ismail et al., 2023; Stojanović et al., 2022). From an empirical perspective, the experiments in Table 4 demonstrate `InterpretCC` 's ability to capture meaningful patterns in the underlying data. The user study addresses whether the outputs of `InterpretCC` explanations are interpretable to users, and can be useful.

# E USER STUDY

In this section, we discuss the details of the user study presented in Section 5.3, discussing the design, content, and additional analyses of the results (including an ANOVA and Tukey HSD tests). We designed the study over four rounds of pilots, with 8 individuals from diverse backgrounds, continuously updating the study design following their feedback. The survey has been approved by the the Human Research Ethics Committee (HREC) under application number HREC 065-2022/27.09.2022.

### E.1 DESIGN OF THE STUDY

We recruit 56 participants using `Prolific`,[4] selecting the ones who identified their current profession as a teacher and who have at least a bachelor's degree. Our target participants have expertise in teaching, as they would be well-suited to understand both the educational context of the study and the consequences of black-box models for student outcomes. During the study, we ask the participants whether they have ever taken or prepared material for an online course (MOOC), their level of education, and what level they are teaching at (from primary school to graduate school). Detailed demographics distribution can be found in Figure 8. The sample of participants is gender-balanced, and about half of them have taken or participated in creating a MOOC. The median completion time is 22 minutes, and the average reward per hour is £14.55.

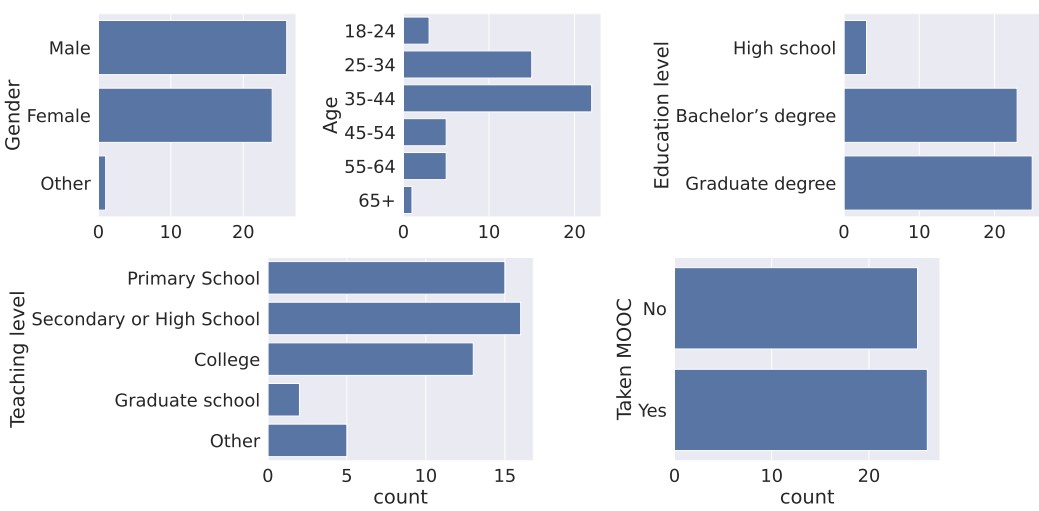

Figure 8: Demographics of teachers that participated in the user study.

At the beginning of the study, the participants are shown the following welcome message and disclaimer:

> Dear participant,
>
> Thank you for participating in our study on model explanations. We are very grateful for your participation and your invaluable insight. Please read this Explanatory Statement in full before moving forward. If you would like further information regarding any aspect of this project, please contact us via the email address provided below.
>
> We are a group of researchers from the [ANON] dedicated to improving education through technology. **The goal of this study is to evaluate different explanations to help a student improve their performance in an online course.**
>
> - This survey has been approved by the [ANON] Human Research Ethics Committee (HREC) under application number HREC 065-2022/27.09.2022. HREC reviews research proposals involving human participants to ensure that they are ethically acceptable.
>
> - All the personal information will be kept confidential and anonymized. Only demographic information is being recorded and will only be reported as aggregate in a way that prevents identification of any individual participant. You can freely withdraw at any time and any collected data you provided so far will be destroyed.

---

[4]www.prolific.com/

- All data will be collected and stored safely and reported in an anonymous form, in accordance with the [ANON] law on data protection ([ANON]).

- Only anonymized or aggregated data may be used in follow-up research (subject to ethics approval), and made available to other researchers for further analysis and for verification of the conclusions reached by the research team.

- Only the principal investigator and the aforementioned researchers have access to the original data under strict confidentiality. Results from the project may be published in conference papers and/or journal articles. In any case, no personal data will be published (only aggregated, anonymous and/or anonymized data will be published).

- Personal data of participants will be stored for 5 years from the date of collection. During this time, participants have the right to access their data and request information about the processing of their personal data. In order to exercise this right, you need to contact the Principal Investigator.

By participating in this survey, you agree that your data can be used for scientific purposes.

**In the following study, you will be asked to compare explanations for approximately 35 minutes.** Please ensure that you have enough time to finish the study correctly. Unfinished or only partially answered studies will not be considered as taken part.

We ask you to approach the questions and exercises with seriousness and to complete them to the best of your ability. We will subsequently check questionnaires for seriousness and will have to discard questionnaires that do not meet this requirement.

Thank you for your help. If you encounter any problem with the survey, or if you want to give extra feedback, or receive additional information, feel free to contact us [ANON].

### E.2 CONTENT OF THE STUDY

First, we explain the setting of the study to the participants with the following introductory message:

You are a teaching assistant helping with a Massive Open Online Course (MOOC). This course is taught at the Master's level with quizzes and video lectures taking place over 10 weeks. Since it's a difficult course with a low passing rate (23.1%), the teaching team wants to help students who are not doing well to perform better in the course by giving them personalized assistance, and encourage students who are already performing well to continue.

To do this, we have models to predict student success or failure using various weekly behavior features, such as number of video clicks or how accurately questions are answered on the weekly quizzes. If potential failure is predicted early (in our case, in the first 4 weeks of the course), we can use the explanation of the prediction to give additional support (i.e. specific tutoring or assignment reminders) to help the student pass the course.

We train four interpretable machine learning models. Each model predicts a student's performance at the end of the course, in the form of "pass" or "fail", but also gives us which factors contribute to student success or failure. We want to compare these explanations according to several criteria:

- Usefulness: This explanation is useful to understand the prediction.
- Trustworthiness: This explanation lets me judge if I should trust the model.
- Actionability: This explanation helps me know how to give feedback to the student.
- Completeness: This explanation has sufficient detail to understand why the prediction was made.
- Conciseness: Every detail of this explanation is necessary.

We randomly sample 4 students from the test set. Among the 4 selected students, 3 failed the class and one passed. For each student, we predict their success or failure with each model and generate an

explanation. We provide them to the participants along with the models' prediction of the student's success or failure. The ground truth (student's performance) and the models' performance are not provided to the participants so that we do not bias their assessment.

The content of the explanation obtained by each method differs greatly. We simplify the explanations and render them in textual and graph format to make them as easy to understand as possible to a human. For `InterpretCC` explanations, we provide the full list of single features / feature groups used by the models. For Feature Gating, we show the evolution of the student's behavior across weeks for each of the features used (see Figure 9). For Group Routing, we compute a generic score for each concept used by the model, by averaging the normalized behavior features that compose the concept. We show the evolution of that concept measure across weeks for the student. We also provide the definition of the concept and of the features that compose it (see Figure 10). For SENN, we select the top 5 groups of students, that we call *concepts*. We showcase all 180 feature-weeks for each concept along with their importance in that concept, highlighting the salient ones. We also provide the importance of each concept for the model's prediction (see Figure 12). Finally, NAM assigns feature importance to all 180 feature-weeks used to make the prediction. We select the 5 feature-weeks found to have highest positive impact, 5 feature weeks with the lowest impact, and the 5 feature-weeks with highest negative impact. We show their importance in a barplot (see Figure 11).

Note that the choice we made for the presentation of the explanations might have an influence on the participants' perception of the explanations. In an ideal setting, we would provide a very detailed description of how each model uses the features and how the explanation is derived, so that the participants can fairly assess the explanation's quality. However, in a realistic setting, the user facing the explanation might not have the time or prior knowledge necessary to understand these elements. To tackle that trade-off between ensuring thoroughness and accessibility of the content of each model's explanation, we opted for a balanced approach in presenting each explanation with a simple graph and an explanatory text that we kept as short as possible.

For each new sample (student taking the course), we provide the list of 4 model explanations in random order. We ask participants to compare these explanations according to five criteria using likert scales. A screenshot of the answer section is shown in Figure 13. We include a practice question to train the participants in how to answer the study and filter inattentive experts. We excluded from the analysis 5 participants who failed to answer correctly to half of the practice questions.

Along with the `InterpretCC` FG graphs, we provide the following explanatory text:

> This student is predicted to fail the course. **The model found the following 2 features to be the most predictive for this student, and only used these features to make the prediction**:
>
> - Quiz Speed on Attempts: The average time passed between two consecutive attempts for the same quiz.
> - Total Time spent on Problems: The total (cumulative) time that a student has spent on problem events.
>
> The plot shows the evolution of the student's behavior for each feature across the 4 weeks.

Along with the `InterpretCC` GR graph, we provide the following explanatory text:

> This student is predicted to fail the course. **For this student, the model decided to only use 1 group(s) of features to make the prediction as this was the one(s) it found most important. The groups of features were designed by expert literature in learning sciences**:
>
> - Concept: PROACTIVITY
>
> Definition: Proactivity measures the extent to which students are on time or ahead of the schedule, as engagement in pre-class activities has shown to be associated with exam performance.
>
> - It includes the following features: The number of videos covered by the student from those that are in subsequent weeks and The average delay in viewing video lectures after they are released to students.

Along with the NAM graph, we provide the following explanatory text:

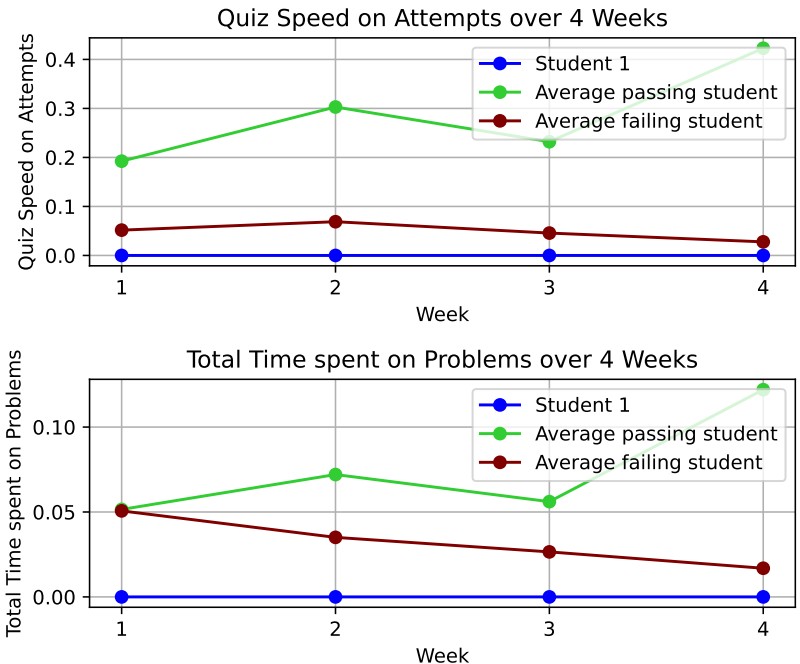

Figure 9: ICC FG method: Importance score visualization of feature-weeks given to the participants for the `InterpretCC` Feature Gating method, for one student.

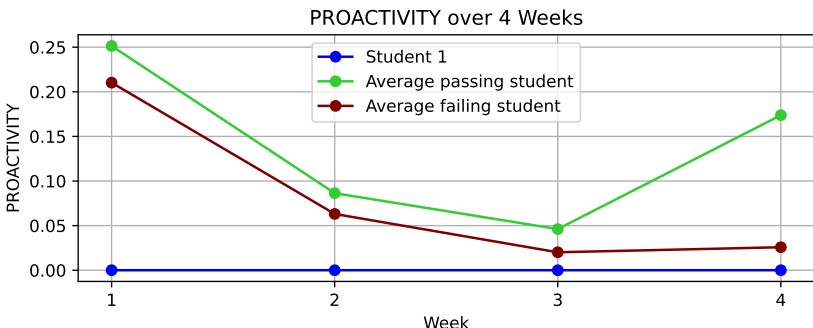

Figure 10: ICC GR method: Importance score visualization of feature-weeks given to the participants for the `InterpretCC` Group Routing with pattern-based matching method, for one student.

> This student is predicted to fail the course. **The model used all 180 feature-weeks (45 features from 4 weeks) to make the prediction. It has assigned a level of importance for each feature-week, showing how much it impacts the predictions, independently of the student's behavior.** Out of the 180 feature-weeks, the plot shows the 5 feature-weeks found to have the highest positive impact, 5 feature weeks with the lowest impact, and the 5 feature-weeks with the highest negative impact. For example, Quiz Max Grade in Few Attempts in week 0 has an importance score of 3.48.

Along with the SENN graph, we provide the following explanatory text:

> This student is predicted to fail the course. **The model used all 180 feature-weeks (45 features from 4 weeks) to make the prediction. It groups them into 5 concepts automatically and assigned a score to each concept.** Each concept can be interpreted as a group of features that

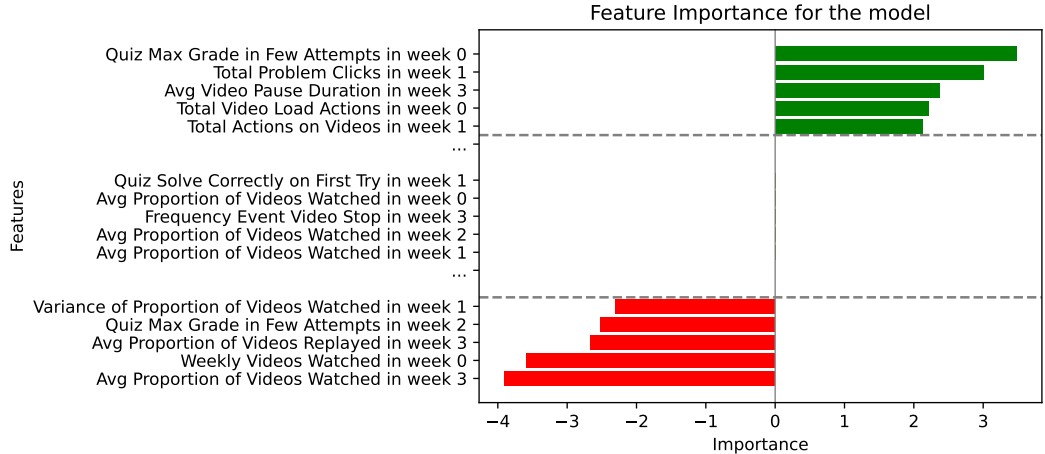

Figure 11: NAM method: Importance score visualization of feature-weeks given to the participants, for one student.

are important for the prediction. The plot shows the importance of each feature-week for each concept, from high positive importance (blue) to high negative importance (red).

Here is the list ordered by absolute value of importance:

- Concept 3 (-0.21)
- Concept 2 (0.08)
- Concept 1 (-0.01)
- Concept 5 (-0.01)
- Concept 4 (0.01)

### E.3 SIGNIFICANCE TESTING FOR USER STUDY RESULTS

We perform an ANOVA to determine the effect of the model and the sample (the student) on the score given by the participants, for each criterion and on average. Table 9 shows the p-values testing the significance of the effect of the model and sample on the participants' scores for each criterion. It can be interpreted the following way. In the first row, if the p-value is lower than the significance level (0.05), then there is a statistically significant difference in scores across the different models for that criterion. In the second row, if the p-value is below 0.05, there is a statistically significant difference in scores across the different students. Finally, a p-value lower than 0.05 in the final row shows that the effect of the model on the scores depends on the student. According to the table, the model has always a significant impact of the value given to each criterion. Then, we apply Tukey's Honest Significant Difference (HSD) Test to determine, for each pair of explanation, if their scores are significantly different (Figure 5).

|  | Usefulness | Trustworthiness | Actionability | Completeness | Conciseness | Global |
|---|---|---|---|---|---|---|
| Model | 0.000 | 0.001 | 0.000 | 0.000 | 0.000 | 0.000 |
| Student | 0.004 | 0.018 | 0.144 | 0.003 | 0.075 | 0.075 |
| Model:Student | 0.143 | 0.058 | 0.063 | 0.000 | 0.171 | 0.171 |

Table 9: ANOVA results for the user study.

Extended description for Figure 5 – Significance test using Tukey's Honest Significant Difference (HSD) Test. It indicates which pairs of models have significantly different means. We highlight the top model on average (*Global satisfaction*, in the last plot), ICC FG (`InterpretCC` Feature Gating) in blue, and the models that are significantly worse according to each criterion in red. Example of

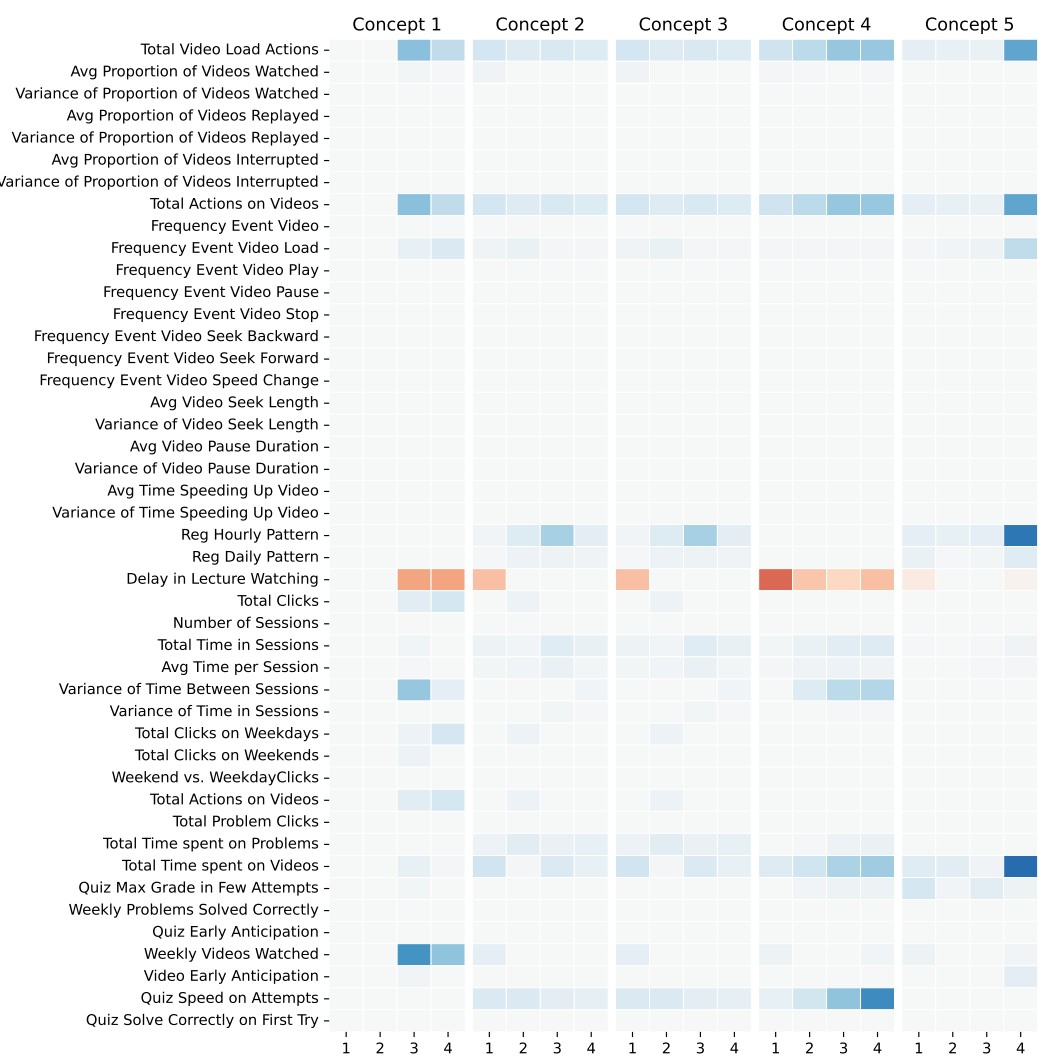

Figure 12: SENN Method: Importance score visualization of feature-week given to the participants, for one student.

Figure 13: Format of the Likert scale question asked for each criterion and explanation.

interpretation using the Tukey HSD test: for the criterion *Usefulness*, ICC FG is scored significantly better than SENN and NAM, but is not significantly better than ICC GR.

## F    REPRODUCIBILITY AND COMPUTE DETAILS

All EDU, Health, and Synthetic experiments were run on a single NVIDIA A100 GPU with 32 GB Memory, each model taking approximately 20 minutes (or often less) to train. Text experiments for InterpretCC took approximately an hour to train. SENN Concepts took the longest time of all models, with each model running within 3-4 hours.

All interpretable models reported in 2 and 3 has been hyperparameter tuned over the following parameters with early-stopping (where applicable):

- learning rate: 1e-3, 1e-4, 2e-5, 1e-5
- layer size: 16, 32, 64
- number of concepts (only for SENN): 5, 6, 7
- batch size: 8, 16, 32, 64)
- gumbel softmax threshold (only for `InterpretCC`): 0.1, 0.3, 0.5, 0.7

Each Feature-Based model was run for 100 epochs with early stopping, and each Concept-Based model was run for 20 epochs with early stopping. Other details related to preprocessing and thresholds are included directly in our repository.

## G    GUMBEL SOFTMAX TRICK AND ITS APPLICATION TO INTERPRETCC

To make the feature gating and routing architectures compatible with backpropagation, we need to make the masks differentiable. These discrete decisions can be trained end-to-end using the Gumbel Softmax trick Jang et al. (2017). This method adapts soft decisions into hard ones while enabling backpropagation, i.e. provides a simple way to draw samples from a categorical distribution.

Given a categorical distribution with class probabilities $\pi = [\pi_1 \, \pi_2 \, ... \, \pi_N]$, one can draw discrete samples $z$ as follows:

$$z = \text{ONEHOT}\left(\arg\max_i [g_i + \log \pi_i]\right)$$

where $g_1...g_N$ are i.i.d. samples drawn from the Gumbel$(0,1)$ distribution. Then, the softmax function is used as a differentiable approximation to $\arg\max$ to generate a $N$-dimensional sample vector $y$ such that

$$y_i = \frac{\exp((\log(\pi_i) + g_i)/\tau)}{\sum_{j=1}^{N} \exp((\log(\pi_j) + g_j)/\tau)} \quad \text{for } i = 1, ..., N$$

where $\tau$ is a softmax temperature parameter that is fixed at $\tau = 1$ for experiments in this project.

Notice that for the gating mechanism, an independent sample is drawn for each 'gate' instead of for each datapoint in routing. For example in feature gating, for each feature $i$, a soft-decision $a_i \in (-\infty, +\infty)$ is outputted by the discriminator layers. The probability $\pi_1$ that the feature should be activated as well as the complement probability $\pi_2$ (feature is not activated) can then be computed by using the sigmoid function:

$$\pi_1 = \sigma(a_i) \qquad \pi_2 = 1 - \pi_1 = 1 - \sigma(a_i)$$

The corresponding (1-dimensional) sample $y$ for each $i$ can thus be reduced to

$$y = \sigma\left(\frac{a_i + g_1 - g_2}{\tau}\right)$$

In other words, the discriminator layers from Fig. 1 actually feed into an adapted Gumbel Sigmoid where $\sigma_i$ is the corresponding $y$ sample as described above.

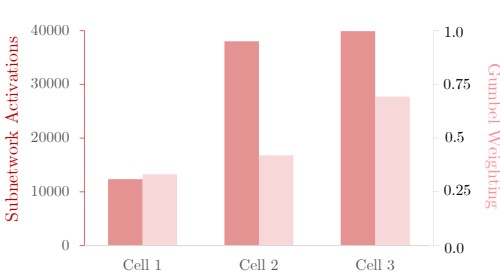

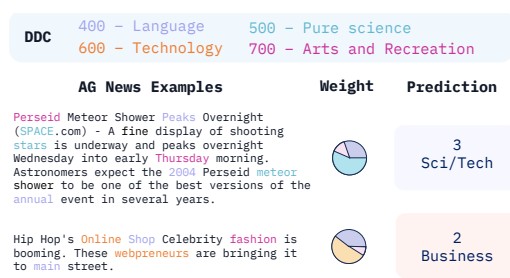

Figure 14: **Breast Cancer**: Number of activations (left) vs. average activation weight (right) per cell. Each subnetwork represents a distinct cell nuclei (10 features).

Figure 15: **AG News**: Two qualitative examples from **AG News** with ICC explanations, where highlighted text represents a subnetwork activation. The pie chart represents the weight of each subnetwork in the resulting prediction.

For routing, the discriminator layers actually output the route logits to a Gumbel Softmax, which constructs the categorical sample vector (of dimension equal to the number of routes and $i$-th entry $y_i$ defined as above).

Finally, we can use a straight-through estimator during training. In other words, binary (or hard/quantized) samples are then used for the forward pass while gradients are obtained from the soft samples for backpropagation. This means that, given soft decisions $\sigma_i$, architectures that use a mask $M = [m_1 \dots m_N]$ with $m_i = \mathbb{1}_{\{\sigma_i \geq 0.5\}}$ differ in value during the forward and backward pass:

$$m_i = \begin{cases} \mathbb{1}_{\{\sigma_i \geq 0.5\}} & \text{(forward pass)}, \\ \sigma_i & \text{(backward pass)} \end{cases}$$

# H    ADDITIONAL GROUP ROUTING EXPERIMENTS

We conduct additional analyses for the high impact, real-world applications in Health, Text, and EDU.

## H.1    GROUPING ANALYSIS FOR BREAST CANCER DATASET

For the **Breast Cancer** data set, the subnetworks grouping features from Cell 1 and Cell 2 are activated much more frequently than the third subnetwork (see Fig. 14). Furthermore, Cell 3 also gets activated with higher weights than the other two cells (Cell 1: 0.25, Cell 2: 0.40, Cell 3: 0.0.70). Smoothness and texture of the tissue images were the most important features across cells.

## H.2    GROUPING ANALYSIS FOR TEXT DATASETS

We provide two illustrative, qualitative examples from the AG News dataset in Figure 15. In the top example (Perseid meteor shower), the words 'stars', 'meteor', and 'SPACE' are routed to the *Pure Science* (500) subnetwork with a 50% activation weight, resulting in the correct prediction of 'Sci/Tech' category. Likewise, for the bottom article, both the *Technology* and *Arts* subnetworks are highly weighted, resulting in the correct prediction of the 'Business' category. Interestingly, subnetwork *Language* (400) is also activated.

## H.3    GROUPING ANALYSIS FOR EDU DATASETS

We conduct sparsity and group routing paradigm analyses on a representative course of the EDU datasets (DSP). We additionally experiment with grouping by paper and pattern over all courses, including two additional MOOC courses with low predictive performance (Structures and Ventures) featured in Swamy et al. (2022a).

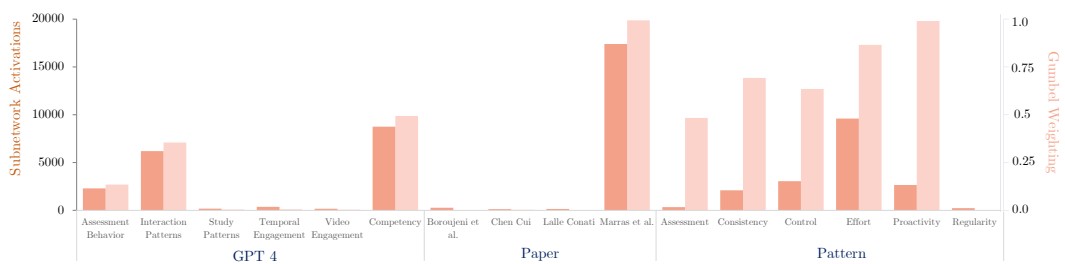

Figure 16: **EDU**: Number of subnetwork activations (left) and Gumbel Softmax activation weights (right) across different groupings (GPT-4, Paper, Pattern) for course DSP 1 of the EDU domain.

### H.3.1 SPARSITY ANALYSIS ON DSP

We conduct a sparsity analysis on the course used for the user study, DSP, and featured in several research works in the learning science community Swamy et al. (2022b;a); Boroujeni et al. (2016). In line with Swamy et al. (2022a), we evaluate the sparsity at both the 40% and 60% prediction setting, representing early prediction at 4 and 6 weeks of the course respectively.

| Setting | Model | Performance | | | Activated Features | |
|---|---|---|---|---|---|---|
| | | ACC | BAC | AUC | $\mu$ | $\sigma$ |
| | Baseline | 0.835 | 0.653 | 0.858 | 97 | 0 |
| | L1 (1e-5) | 0.790 | 0.711 | 0.817 | 8.54 | 0.58 |
| 40% | L1 (1e-4) | 0.763 | 0.748 | 0.784 | 2.20 | 1.64 |
| | Annealed MSE | 0.768 | 0.770 | 0.823 | 13.26 | 3.72 |
| | Truncated AMSE | 0.805 | 0.743 | 0.823 | 5.95 | 1.03 |
| | Baseline | 0.944 | 0.925 | 0.982 | 97 | 0 |
| | L1 (1e-5) | 0.914 | 0.935 | 0.963 | 37.60 | 3.97 |
| 60% | L1 (1e-4) | 0.914 | 0.917 | 0.957 | 31.38 | 4.45 |
| | Annealed L1 (1e-5) | 0.910 | 0.927 | 0.957 | 38.36 | 3.61 |
| | Annealed MSE | 0.892 | 0.927 | 0.952 | 21.45 | 3.23 |
| | Truncated AMSE | 0.787 | 0.788 | 0.839 | - | - |

Table 10: **InterpretCC Feature Gating** comparison of performance metrics between different sparsification methods next to their average and standard deviation of number of activated features for both 40% and 60% early success prediction settings; baseline benchmarks also provided for contrast.

An annealed mean-squared regularization proved most effective, although it activated more features on average than $L_1$-norm regularization which more effectively reduced the feature space while achieving desirable balanced accuracy performance in this setting. By truncating the initial feature space to only the activated features and using the same architecture, performance is almost maintained even though the average number of activated features per datapoint is more than halved (from around 13 to 6). However, this method was not as effective for the 60% setting. Truncating the feature space largely reduces predictive capability (e.g. almost a 0.15 drop in balanced accuracy). L1 regularization in this case proved best. Using an annealed regularization did not significantly improve or change model performance as well.

### H.3.2 MULTIPLE GROUPING PARADIGMS ON DSP

To illustrate the influence of different feature groupings, we conduct a deep dive for course DSP 1 of the EDU domain. Figure 16 illustrates the number of subnetwork activations and corresponding weights for three different groupings.

For the first two groupings (GPT-4, Paper), the subnetwork activations (number of times the route was activated) closely mirror the Gumbel Sigmoid adaptive weighting, indicating that a few networks

are activated with high weights for prediction. In the *group by GPT-4* setting, we see behaviors of competency, interaction patterns, and assessment frequently activated for student pass-fail predictions. Although 'interaction patterns' is the largest category (most number of features chosen by GPT-4), it still comes second to competency (focusing on student achievement). In the *group by paper* setting, we see a clear preference for Marras et al. with over $17,500$ students predicted using this network (dark orange) and high weight given to the predictions from the network (light orange).

In contrast, in the third grouping (Pattern), we see a differentiation between the number of activations (dark orange) and the weight of the activations (light orange). Notably, the patterns of 'Effort', 'Proactivity', 'Consistency' and 'Control' all have higher than $50\%$ weight when they are activated, which means they contribute a lot to the overall prediction when chosen.

### H.3.3 GROUPING BY PAPER AND PATTERN OVER ALL COURSES

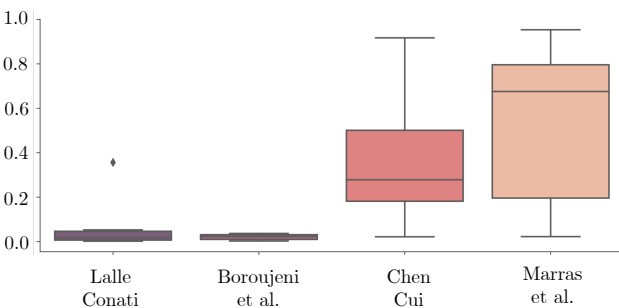

Figure 17: **EDU** analysis of *group by paper* routing averaged over all six courses for each paper grouping. The y-axis represents the proportion of points for which the subnetwork is activated.

In Figure 17, we see `InterpretCC` routing by research paper (grouping the features based on the paper they were proposed in). The Marras et al. and Chen Cui feature sets have clearly been identified as important over the majority of courses, echoing findings in other learning science literature using BiLSTM and random forest architectures Marras et al. (2021); Chen and Cui (2020); Swamy et al. (2023a). The large standard deviations in the box-plots indicate that for at least some courses (in this case Structures and Venture), Chen Cui and Marras were not found significantly important. Notably, the same courses that have low accuracies on routing in 2 are those that have low scores on the two most popular feature sets, showing a consensus among performant `InterpretCC` models and a validation of the identification of importance.

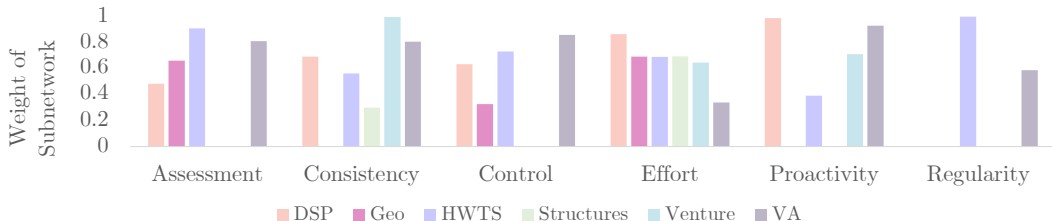

Figure 18: **EDU** Weighting distribution across subnetworks using *group by pattern* for 6 courses.

In Fig. 18, we see a widely varying distribution of patterns selected across courses, showcasing the ability of `InterpretCC` to adaptively select subnetwork weights depending on the dataset.

## I INTERPRETCC'S RELATIONSHIP WITH INTERPRETABILITY

We discuss a comparison of a traditionally interpretable machine learning model (random forest) with `InterpretCC`, and we propose an architecture extension to make `InterpretCC` even more interpretable (at the expense of additional complexity).

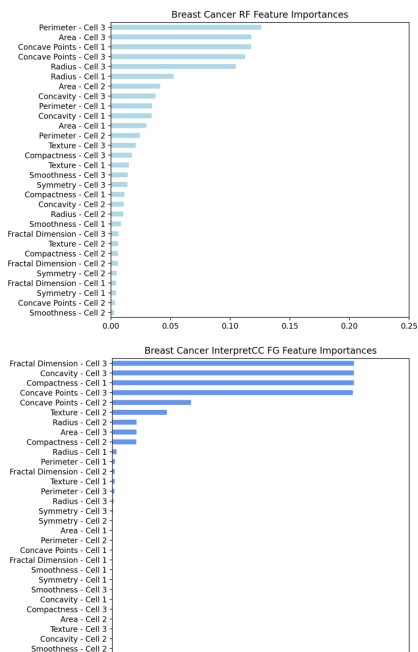

| Model | B. Cancer | Synthetic |
|---|---|---|
| Random Forest (RF) | *92.98 ± 4.70* | 85.32 ± 3.48 |
| Non-interpretable base module (NN) (SHAP, LIME) | 89.70 ± 1.05 | 86.67 ± 0.31 |
| NAM | 88.77 ± 7.31 | 87.85 ± 1.31 |
| SENN Features | 80.52 ± 6.21 | 83.67 ± 1.86 |
| SENN Concepts | 85.26 ± 1.03 | *89.51 ± 0.51* |
| InterpretCC FG | 78.19 ± 3.54 | 84.67 ± 4.04 |
| InterpretCC Top K | 84.66 ± 3.02 | **90.83 ± 1.93** |
| InterpretCC GR | **94.85 ± 1.25** | 89.47 ± 2.89 |

Figure 19: **InterpretCC vs. RF Tabular Comparison**. (left) Performance of models on Breast Cancer and Synthetic datasets. (right) Features chosen in explanations from RF (light blue) vs. InterpretCC for Breast Cancer dataset (dark blue).

## I.1 INTERPRETCC TABULAR COMPARISON WITH A TRADITIONAL INTERPRETABLE MODEL

In Fig. 19, the table (left) shows that all models, in comparison with a Random Forest model (RF), overlap in 95% CI. The performance of a tuned RF is in line with deep learning approaches. However, the RF uses 100 trees with average depth 7.37 +/- 1.16 for Breast Cancer and 1000 trees with average depth 6.21 +/- 2.03 for Synthetic data, so is not simple to understand the decision path.

We then compare which features RF selects as important (light blue) vs. which features ICC FG selects as important (dark blue): ICC's feature importances are much sparser.

## I.2 INTERPRETCC DESCRIPTION OF FAITHFULNESS

As inspired from the OpenXAI benchmark (Agarwal et al., 2022):

- **Feature Agreement (FA):** Computes the fraction of top-K features shared between an explanation and [the underlying data pattern].
- **Rank Agreement (RA):** Measures the fraction of top-K features that are not only shared but also appear in the same rank order in both the explanation and the [underlying data pattern].

We define **Ground Truth Alignment** as the fraction of features shared between an explanation and the underlying data pattern (K=total number of features). We define **Ground Truth Faithfulness** as the fraction of features shared between an explanation and the ground truth explanation (exactly what the model is using as a decision process).

As an extension of the results in Table 4, we conduct a Pairwise Rank Agreement analysis. **Pairwise Rank Agreement (PRA)** assesses whether the relative ordering of feature pairs is consistent between the explanation and underlying data pattern, computing the fraction of pairs with the same relative importance. Comparing the eight models on the synthetic data for PRA, the results are: 69.71 +/- 3.89 (ICC FG), 74.55 +/- 1.21 (ICC GR), 71.99 +/- 3.25 (SENN Features), 45.30 +/- 7.49 (SENN Concepts), 70.16 +/- 5.63 (NAM), 82.13 +/- 7.21 (IG), 73.31 +/- 1.91 (LIME), and 72.93 +/- 3.71 (SHAP). As this metric is simply a less-strict version of Rank Agreement and the order of important

features is very important in downstream human-centric tasks deriving from explanations, it is not included in the main results. Notably, all 95% CI overlap again, except for ICC GR and SENN Concepts, where SENN Concepts is significantly worse than ICC GR.

## I.3    INTERPRETCC EXTENSION FOR AN INTERPRETABLE DISCRIMINATOR NETWORK

`InterpretCC` can enable an interpretable discriminator network simply by using a set of decision trees (or SVMs, LRs, any other traditionally interpretable model) instead of a neural network (FG: one model per feature or GR: one model per concept). This additional interpretability would come at the expense of model/explanation complexity and perhaps reduced accuracy as it doesn't take into account cross-feature interactions, but would allow us to make statements like: "Concept X was chosen because feature A > value, value > B < value, C > value, D = 0, and E > value. Concept X contributes 35% to the prediction."

**Concrete Implementation Details**:  For `InterpretCC` FG, each tree would predict a binary decision between "keep this feature" or "don't keep this feature" with the input of the full feature space. For `InterpretCC` GR, each SVM would assign a score between 0 and 1 regarding whether to "keep this concept" with the input of the full feature space (the scores would then be normalized across concepts/features). These would be trained together with the predictive network and a sparsity criterion in exactly that same way as we have shown with `InterpretCC`: all that changes is the choice of which model to use for the discriminator network. A simpler NN solution is to add an attention layer to the discriminator for some opaque interpretability.

However, we do not include this approach in the model as it makes both the explanation for the downstream user more complex and the model itself more complex; we choose to focus instead on the design criteria of optimizing human actionability through sparsity and simplicity.

## I.4    INTERPRETCC COMPARISON TO EXTRACTIVE RATIONALE METHODS

In comparison to InterpretCC, extractive-rationale methods from the NLP community like FRESH (Jain et al. 2020) provide explanations that are more tailored for the text domain. However, these methods have several weaknesses, illustrated in a comparison of FRESH to InterpretCC: 1) These methods are often harder to generalize to new tasks, e.g. FRESH needs to train 3 new models sequentially instead of 1 model with two parts in parallel for every new setting. 2) There is often bias included in determining the initial importance or saliency scores (LIME, Attention, Gradients used in FRESH) instead of letting the model learn directly. 3) These models are longer, larger, and more complex to train over InterpretCC's simpler architectures. 4) Lastly, and most importantly, methods like FRESH require the selection of a contiguous section of text as "rationale", in contrast to InterpretCC's mapping of words to concepts from the Dewey Decimal System (or any other grouping methodology). While this contiguous selection makes sense in the text domain, for tabular, or even time-series (beyond anomaly detection) data, this kind of explanation is not suitable or human-friendly. Therefore, we did not implement FRESH on the other modality experiments as it would be fundamentally unsuited to the data format.

For quantitative experiments (showcased in Table 2, we tuned the FRESH model over rationale length and learning rate, and ran experiments with 5 random seeds with the exact architecture used in the paper. Notably, FRESH experiments use BERT models, while ICC experiments use the smaller DistillBERT model. For SST, FRESH uses a 30% rationale size with an average accuracy of 82.05% +/- 0.56%, where ICC FG has 88.21% +/- 3.41%, ICC TopK has 92.98% +/- 0.88%, and ICC GR has 91.75% +/- 1.86%, which are significantly more performant. As SST has 7 words on average, ICC FG selects a larger explanation size, on average 3 or 4 words (indicated in Figure 2), but maintains a higher accuracy. For AG News, FRESH uses a 20% average rational length, with performance of 88.73% +/- 0.69%. This is comparable to the ICC GR numbers 90.35% +/- 1.07% and the the ICC Top K routing approach with 87.25% +/- 2.48%, and more performant than the ICC FG approach 85.72% +/- 5.31% (although the high variation in the FG approach is to be noted), and the 95% CIs overlap showing that the approaches are similar.

Other methods from the extractive-rationale community, like the methods proposed by (Bastings et al., 2019), (Yu et al., 2019), and `InterpretCC` take fundamentally different approaches to model interpretability, each with distinct trade-offs. Bastings et al. focus on token-level rationale selection

using binary gates optimized through REINFORCE, which allows for precise, fine-grained explanations but suffers from optimization challenges and limited applicability beyond text-based tasks. Yu et al. extend this idea by introducing a generator-predictor framework that enforces complementarity between selected and unselected features. While this ensures higher rationale quality, it also significantly increases model complexity and computational cost. `InterpretCC` diverges by leveraging human-defined feature groups and a unified, end-to-end architecture, balancing interpretability and scalability. Unlike token-based approaches, `InterpretCC` focuses on concept-level explanations that generalize across domains like tabular and time-series data, enabling more actionable insights. By emphasizing cross-feature interactions and the integration of domain knowledge, `InterpretCC` differs from approaches in the extractive-rationale community.

| Aspect | Bastings et al. (2019) | Yu et al. (2019) | Jain et al. (2020) | InterpretCC |
|---|---|---|---|---|
| **Feature Selection** | Token-level binary gates | Token-level generator-predictor | Contiguous text spans as rationales | Feature-level binary gates or group-level selection (domain-informed) |
| **Architecture** | REINFORCE optimization | Generator + predictor (cooperative) | Generator + predictor (extractive rationales) | Single end-to-end model with routing |
| **Domain Focus** | Text-focused | Text-focused | Text-focused | General (text, tabular, time-series) |
| **Explanation Type** | Token rationales | Token rationales with complementarity | Contiguous extractive rationales | Concept-based, human-centric rationales |
| **Optimization Complexity** | High due to discrete variables | Moderate with generator constraints | Moderate due to sequential training | Low with Gumbel-Softmax routing |
| **Main Strength** | Fine-grained token selection | Explicitly enforces rationale quality | Faithfulness to predictions | Human-friendly explanations |
| **Main Weakness** | Optimization variance; text-only | High model complexity; text-only | Task-specific rationales; text-only | Requires interpretable feature group design (human or LLM) |

Table 11: Design comparison of `InterpretCC` to Extractive Rationale Methods.

## I.5 INTERPRETCC EXTENSION FOR CROSS-FEATURE OR RAW MODALITY INTERACTIONS

Graph-based models could be very useful in the `InterpretCC` architecture for raw time series data, with a natural extension in the discriminator stage. The message passing graph network and concept activation vector approach showcased with RIPPLE (Asadi et al., 2023) could be used to define a sparse adjacency matrix as opposed to a sparse vector on the input. This could then relate to interaction-based subnetworks, referring to multiple features in each concept used in the explanation. Alternatively, graph models like RAINDROP (Zhang et al.) or SGP (Cini et al., 2023) could be used in the predictive module stage, where each subnetwork is a graph model only focused on the specific modalities or features passed into the subnetwork. Using the sparseness enabled by the discriminator layers, only a few of the graph models would be activated for each point's prediction.

Without this extension, it is possible `InterpretCC` could be used directly on raw time series data to simply specifying the "concepts" as fixed or relative time intervals (analogous to anomaly detection), enabling users to identify which parts of the time-series were used in the prediction. This could answer questions like: "Was my behavior in week 1 useful in predicting that I performed well on the exam?"). This significant time interval approach could be further extended to modalities like raw speech or video, although using modality-specific expert grouping methodologies would be preferred, as this would be more useful for downstream user actions.