# OpenReview forum: "Intrinsic User-Centric Interpretability through Global Mixture of Experts"
_ICLR.cc/2025/Conference — ICLR 2025 Poster_

### Official Review · Reviewer_WueB · 2024-10-26

**Soundness:** 2
**Presentation:** 3
**Contribution:** 2
**Rating:** 6
**Confidence:** 3

**Summary:**

This paper introduce a set of intrinsically (by design) interpretable machine learning methods, motivated by a global mixture of experts method. The goal of these methods (InterpretCC) focus on accurate predictions and aims to provide faithful explanations. The authors conduct experiments on a variety of tasks, including some user studies, and demonstrated the effectiveness of their approach.

**Strengths:**

1. the motivation of this work is clear and solid, and I believe intrinsic explainable machine learning should definitely be an important direction of research with great importance
2. good amount of experiments are conducted (including time series, text and tabular inputs) and the experimental results seems to be reasonable
3. the proposed methods seems to be reasonable novel (based on a global mixture of experts, with non-overlapping group/ set selections)

**Weaknesses:**

1. there exist evidence of over-claiming (such as Table 1, method comparison; after reading the paper, I am not convinced that the proposed methods actually achieve "faithfulness", and maybe somewhat allow full "coverage") and definition of these terms are also not clear / or there is no solid definition (although briefly mentioned in background -- Interpretability Foundations)

2. I am fully understand that explanation methods like InterpretCC are build based on expert knowledge or domain expertise (based on l231-296), are other intrinsic baseline methods adopt the same concepts/expert knowledge? would this really be a fair comparison otherwise? Also, for explainable machine learning, there is a trade-off between injecting how much domain knowledge v.s. black box methods (such as LIME or SHARP), could you add a discussion on if such domain knowledge does not exist (as this can often be the case in real-life scenarios)?

**Questions:**

Please refers to my questions in the "weakness section". The general landscape of explainable machine learning requires a stronger and more scientific taxonomy / definitions of these terms, the authors are thus encourage to discuss / elaborate on this with respect to their proposed methods.

I think this paper also lack a bit of theoretical analysis/justification (or at least tried to explain why the global mixture of experts method make sense and why it works for all these tasks). Could the authors try to discuss this?

---

> ### Author Response · Authors · 2024-11-24
>
> We kindly thank Reviewer 4 for this careful evaluation of our work and greatly appreciate the review, highlighting our paper’s strengths in presentation, “novel” methods, and “clear and solid” motivation. We strongly agree with all points raised.
>
> We have addressed each in a point-by-point response below and feel that the resulting edits have further improved the manuscript.
>
> **Potential over-claiming due to unclear definitions of certain axes of approach comparisons.** We thank the reviewer for the suggestion that will greatly improve the clarity of our paper’s contribution. We have now included a full taxonomy of terms based on definitions of explanation criteria derived from related literature, which have been included in Appendix A. We have also clarified the claims in the introduction and tied in more literature in the “Interpretability Foundations” section.
>
> The reviewer also asks for justification of our categorization of “faithfulness” and “coverage” in light of the InterpretCC approaches. The definition of explanation faithfulness is the “alignment between the explanation and the model’s inherent decision making process (ground truth)” (Lyu et al. 2022, Dasgupta et al. 2022). As the explanation in InterpretCC FG (selected features from the discriminator layer) and InterpretCC GR (selected concepts from the discriminator layer) are directly used in the prediction of the instance, we believe this makes the explanation faithful to the model’s decision making process. The definition of explanation coverage is the “the extent to which an explanation accounts for the features or concepts utilized by a model to make a prediction” (Ali et al. 2023). Both InterpretCC FG and GR provide explanations that explicitly select a subset of the feature space, and only use the features that are selected in the prediction. Therefore, both models have full coverage (the explanation covers all the features used in the model’s decision process).
>
> **Comparison to other intrinsic methods that use expert knowledge.** After a thorough literature review, we do not find a similar intrinsic approach that uses expert knowledge to separate the fully separate feature space into concepts. There are a few models, primarily in the vision domain, where concepts are defined through expert-selected prototype examples or rules (Koh et al. 2020, Konstantinov et al. 2024). We add these to the discussion in Sec. 2. A stronger comparison point is to expert-specified in-hoc approaches (i.e. concept activation vectors) that have been discussed already in the background section.
>
> Regarding R4’s concern of fair comparison, all explainability approaches we compare ICC to (i.e. SENN, NAM, LIME, SHAP, IG, TCAV, and the newly added FRESH) use the exact same input features and modeling setting as ICC with real world use cases, creating a fair comparison between both intrinsic and post-hoc approaches. If anything, methods like SENN Concepts, D-TCAV, or Sum-of-Parts which optimize the selection of feature groupings would intuitively have a performance advantage over expert-specified knowledge, as ICC’s grouping is centered on user actionability and usefulness instead of optimizing directly for predictive potential.
>
> **Settings where domain knowledge cannot be easily found.** Indeed, our method requires specification domain knowledge, i.e. expert input. In settings like the breast cancer experiments, clinical knowledge that is prediction task-specific is required to interpret the results of the tests into even more meaningful medical categories, although we were able to group by cell and by lab test. However, LLMs continue to get better at grouping features into concepts, as evidenced by our experiment in Table 4 with a GPT 4 model, where without expert feedback, concepts that were selected were both human-understandable and effective at prediction in comparison to expert-extracted categories. We believe the abilities of LLMs to group descriptions of input features or raw input streams into meaningful concepts will only increase the effectiveness of the ICC models. We have added a nuanced discussion of this into work in Sec. 6.
>
> Additionally, for modalities like raw time series, it is very hard to have human interpretable knowledge. Graph based models could be very useful in the InterpretCC architecture for raw time series data, with a natural extension in the discriminator stage. The message passing graph network and concept activation vector approach showcased with RIPPLE (Asadi et al. 2023) could be used to define a sparse adjacency matrix as opposed to a sparse vector on the input. This could then relate to interaction-based subnetworks, referring to multiple features in each concept used in the explanation.
>
> [continued in the next comment]

---

> > ### Author Response · Authors · 2024-11-24
> >
> > Alternatively, graph models like RAINDROP (Zhang et al. 2022) or SGP (Cini et al. 2023) could be used in the predictive module stage, where each subnetwork is a graph model only focused on the specific modalities or features passed into the subnetwork. Using the sparseness enabled by the discriminator layers, only a few of the graph models would be activated for each point’s prediction. We have added this discussion of InterpretCC’s extension to graph models in Appendix I.5.
> >
> > Lastly, our grouping approach and integration of domain knowledge are directly focused on actionability. If it is not possible to use expert or LLM knowledge to separate the feature space into concepts, then it is also might not be useful to interpret these concepts into actions that have downstream consequences.
> >
> > **Additional theoretical analysis and intuition behind the design of the InterpretCC FG and GR model.** For the GR model, we believe the specialization of subnetworks in our global mixture of experts model is directly crucial to our approach. Limiting the feature space for each subnetwork to a smaller subset of features allows the subnetwork to learn more granular and specific patterns. Leveraging multiple specialist subnetworks together for the prediction is intuitively better than one monolithic network that might not fully be able to capture all of the underlying patterns.
> >
> > Similarly, the expert knowledge of which features should be considered together can be helpful for the model to not over-focus on spurious correlations in the training setting that do not translate to inference time. For InterpretCC FG, the quality of predictions improves due to the focus on a few input features, as well as the ability to adaptively select as few features as necessary, creating both a reduction of noise, and optimizing the interpretability - accuracy tradeoff through sparsity.
> >
> > The human-centered grouping methodology is directly confirmed by the usefulness of the explanations, and demonstrated in an initial user study, although we strongly encourage the community to examine many different real-world use cases. We have now added these points to Sec. 6.
> >
> > We thank the reviewer for their feedback and valuable suggestions and questions. We hope that all of their concerns have been addressed, and we feel that the paper is much improved.

---

### Official Review · Reviewer_CDui · 2024-11-02

**Soundness:** 3
**Presentation:** 4
**Contribution:** 3
**Rating:** 8
**Confidence:** 3

**Summary:**

This paper proposed the intrinsic explanation framework called InterpretCC to combine feature gating and group routing based on the design space by considering user-centric features to enable actionable interpretation without sacrificing prediction performance compared with traditional prediction models. It conducted user evaluation and extended intrinsic explanation models to other less popular modalities, such as time series, tabular and text.

**Strengths:**

The unique strength of this paper is to put user-centric design into intrinsic explanation models, aiming to address the applicability of developed models in general domains. It is nice to see that efforts of pushing model design into user-centric design.

It extends the intrinsic explanation models from visions to other modalities, especially time series, text and tabular datasets.

It has conducted systematic evaluation of four domains with their baseline different explanation models and measured the interpretation based on several well-defined metrics.

The user evaluation of recruiting 56 teachers to judge the usefulness of the interpretation is a great plus.

**Weaknesses:**

- In real world scenarios, explanations are not just a set of features, rather than the interactions of a pair of features. Do you consider to identify the interactions of features in your interpretCC framemwork
- In your user evaluation, as your method is providing local interpretation, as mentioned that interpretCC can recommend interpretation like "“This student was predicted to pass the course because and only because of the student’s regularity and video watching behavior". How can you prove such if and only if situation, because all prediction methods in interpretCC are association based, not causal based.
- can you please provide more details on how to deal with the grouping of features (it is great to see that you are using LLM to group features) for MoE, but grouping features itself can be challenge as some features might belong to two different groups, which could lead to errors in feature gating, and MoE.

**Questions:**

- can you think about using graph model to extend the explanability from single feature based to feature interaction based.
- in your evaluation, can you provide more details about features in EDU example, in Figure 10 in appendix, many features are similar, how these teachers could look at these features to measure whether "“This student was predicted to pass the course because and only because of the student’s regularity and video watching behavior".
- can you please provide more details on how the user predefined features (or user preferred features) are considered in feature gating and group routing.

---

> ### Author Response · Authors · 2024-11-24
>
> We kindly thank Reviewer 3 for the careful evaluation of our work and greatly appreciate the strongly positive review, highlighting our presentation, soundness and contribution, as well as the “great plus” of the user evaluation. We agree with all points raised.
>
> We have addressed each in a point-by-point response below and feel that the resulting edits have further improved the manuscript.
>
> **Interaction of features is important for explanations in real world settings.** We agree, we discuss this more now in Appendix I.5. With the IntepretCC Group Routing model, the design of concepts inherently utilize the shared interactions of related features in the predictive subnetworks. Similarly, for the Feature Gating case, selecting the features first and then having a predictive model see all the “important” features together is an advantage that takes into account the features’ shared predictive potential. This is in contrast to other intrinsic approaches like NAM or SENN, which assign linear scores to each feature individually. While InterpretCC FG selects the important features automatically (and predicts based on their shared interactions), InterpretCC GR uses user-specified or LLM-specified concepts, creating explanations with shared feature interactions that are by-design actionable for the user. We believe in expert-specified concepts because if the explanation is not actionable, it might not be useful for anything other than model auditing.
>
> **Rephrasing of the “because and only because” terminology used to describe InterpretCC explanations, due to association and not causal claims.** It is possible (rare case) that the discriminator selects regularity and video watching but does not fully use them in the prediction. The model is trained to explicitly avoid this behavior (harsh sparsity constraints, minimizing predictive losses), but it is still possible. As per the reviewer’s suggestion, we have revised the sentence on line 87 to be: "The student’s regularity and video watching behavior were the only two aspects selected as important for the student's prediction of passing the course, and the model did not use any other aspects to make this prediction."
>
> **Additional details on how to group features in challenging cases (i.e. when a certain feature could belong in two groups).** Great suggestion, we have incorporated a discussion of this in Appendix C. In InterpretCC, we design towards the user's actionability of the resulting explanation. Therefore, if the knowledge that a feature is important can lead to a specific action, and if this action is the same one that should be taken for other features, then those features should be grouped together. From the modeling perspective, grouping features together in a concept means that their shared predictive potential should be leveraged, and likely this is more important for one type of features than another. We maintain that a feature should not be placed in multiple groups to have the faithfulness guarantees that are a strength of InterpretCC. However, in a rare and specific case where two actions must be taken based on the feature, or it is too difficult to decide which concept the feature belongs in, it is always possible to 1) put that feature in its own subnetwork (and have it be selected alongside any of the other feature groups), or 2) combine the two subnetworks into a larger concept.
>
> **Possibility to use graph models to extend the explainability from single feature based to feature interaction based.** Yes, definitely. Graph based models can be very useful in the InterpretCC architecture, as a natural extension in the discriminator stage. For example, the message passing graph network and concept activation vector approach showcased in RIPPLE (Asadi et al. 2023) could be used to define a sparse adjacency matrix as opposed to a sparse vector on the input. This could then relate to interaction-based subnetworks, referring to multiple features in each concept used in the explanation. Alternatively, graph models like Zhang et al. 2022 could be used in the predictive module stage, where each subnetwork is a graph model only focused on the specific modalities or features passed into the subnetwork. Using the sparseness enabled by the discriminator layers, only a few of the graph models would be activated for each point’s prediction. We have added this discussion of InterpretCC’s extension to graph models into Appendix I.5.
>
> [continued in next comment]

---

> > ### Author Response · Authors · 2024-11-24
> >
> > **More details about the features in the EDU example, especially regarding Fig. 10 in the Appendix and teachers’ interpretation of it.** Figure 10 shows the visualization that was given to the user study participants for the SENN method, for one student. Along with this visualization, we provided the following explanatory text: “[...] The model used all 180 feature-weeks (45 features from 4 weeks) to make the prediction. It groups them into 5 concepts automatically and assigned a score to each concept. Each concept can be interpreted as a group of features that are important for the prediction. [...]”. The concepts provided by SENN are not immediately interpretable into human-understandable, coherent feature groups in the modality of time series data. Explanations such as “This student was predicted to pass the course because and only because of the student’s regularity and video watching behavior” can only be obtained using expert-created interpretable feature grouping defined in Table 7 in Appendix C, for the InterpretCC Group Routing models. These feature groups are used by InterpretCC Group Routing, whose visualization for the user study can be found in Figure 8. We have now described this more clearly in Appendix.
> >
> > **More details on how the user predefined features are considered in feature gating and group routing.** This is an important clarification. Users specify a grouping that separates individual features into concepts (which are groups of features), specifically dealing with the routing aspect of ICC GR. User selected features and user definitions of features are not required in ICC FG. For concept specification, users describe feature groupings in the form of a list or dictionary with which features are assigned to which network. In training, the discriminator takes in all the features as input and decides which of the N concepts to activate. In prediction, there are N predictive modules (one for each concept), each with only the features that are mapped to that concept as input. If a specific concept is activated, then only the features assigned to that concept (by a user) are sent to that subnetwork, and this is used for the eventual prediction. We have now described this more clearly in Sec. 3 and Appendix C of the main paper.
> >
> > We thank the reviewer for their strong, positive feedback and valuable questions. We hope that all of their suggestions have been addressed.

---

> > ### Comment · Reviewer_CDui · 2024-11-26
> >
> > Thank authors for addressing my concerns. I am satisfied with the answers.

---

> > > ### Author Response · Authors · 2024-11-28
> > >
> > > We thank R3 for reviewing our response. We are grateful to hear that all concerns have been addressed!

---

### Official Review · Reviewer_csPd · 2024-11-06

**Soundness:** 3
**Presentation:** 4
**Contribution:** 3
**Rating:** 8
**Confidence:** 4

**Summary:**

The authors propose two new neural network architectures that are designed to be intrinsically interpretable. The first uses a gating mechanism to select a sparse set of features, and the second uses a mixture-of-experts approach to select a sparse set of interpretable feature groups. In experiments on five datasets spanning three modalities, the authors show that their method achieves comparable (or better) performance compared to black-box, non-interpretable models and two intrinsically interpretable baselines. The authors also conduct a user study involving teachers and the task of predicting student performance, and they find that the explanations produced with their method are preferred compared to baselines.

**Strengths:**

1. **Simple, intuitive, and novel idea for the design of intrinsically interpretable neural network architectures.** The authors present a novel idea for the design of neural network-based models that are intrinsically interpretable yet retain strong performance.
2. **Thorough experimental analysis.** The authors evaluate their method on five datasets, spanning several different domains and modalities. They analyze both the performance of the model and the quality of the explanations it produces in terms of faithfulness, sparsity, and human satisfaction.
3. **Results show that the proposed method produces useful explanations without sacrificing performance.** Across the five datasets, the proposed method performs comparably (or better) to black-box models with the same architectures in terms of predictive accuracy. The authors also show that the explanations produced by the proposed method tend to be sparse in terms of the percentage of features that are activated. The explanations also achieve high faithfulness scores on the synthetic data. Finally, the authors conduct a user study with 56 teachers, where they apply their method to the task of predicting student performance. They find that the study participants prefer the explanations produced by their method compared to baselines in terms of almost all of the criteria examined (e.g, usefulness, trustworthiness) as well as overall.
4. **Writing clarity.** The paper is clearly written and easy to follow.

**Weaknesses:**

1. **Missing comparison/discussion of prior work on explain-then-predict / extractive rationale methods.** There is a substantial amount of existing work on intrinsically interpretable models that involve the same basic two steps proposed in this work: (1) select a subset of the input as the “explanation”/”rationale” and (2) use a model that sees only this explanation to make the final prediction. A lot of this has been done in the NLP space; see the discussion in Section 4.5.2 in [1], and the specific methods in [2]-[5]. Since these works take the same basic approach to producing explanations, I think they should be included as baselines in the evaluation. At the very least, the authors should mention this work in the related work section and justify why their work is sufficiently different such that an experimental comparison is not needed. As a related point, the authors say in their intro that prior work on intrinsically explainable models is “rare” for “text modalities”, and they say that one of their contributions is extending intrinsic interpretability methods to “modalities and domains” that are less common for this area, such as text. I’m not entirely convinced by this point, especially since the authors did not mention any existing intrinsically interpretable approaches for text data (e.g., [2]-[5]) in their related work section.
2. **Primarily applicable to cases in which model inputs are composed of interpretable features.** The explanations produced by the proposed method take the form of a subset of model inputs (and in the group routing version, a subset with group labels). While this is human-understandable in the case in which model inputs are human-understandable (e.g., time-series features or words in a document), it is not clear that the explanations would be useful in cases where the model inputs are less structured/interpretable (e.g., pixels in an image, raw time-series data, text tokens). In many applications, the most performant models use raw/complex data as inputs as opposed to handcrafted features. Therefore, this seems to be a major limitation of the method. And it is not discussed in the paper. In addition, all experiments in the paper involve model inputs that consist of interpretable features (i.e., words, handcrafted times-series features, or image features). I would like to understand to what extent the method can be applied when the inputs are images, raw time-series, speech, text tokens, etc.
3. **Doesn’t address the possibility that explanations produced with their method are misleading/unfaithful.** Although the authors claim that their method is guaranteed to be faithful, I don’t think this is actually the case. As pointed out in prior work (e.g., [6], [7]), “select-then-predict” methods of this nature can produce misleading explanations. For example, it could be the case that the predictive model looks for superficial patterns in the selected feature set (e.g., how many features are selected) rather than uses the features as a human would expect. The authors do not address this risk in their paper.
4. **Limited analysis of impact of sparsity threshold.** In Section 6, the authors state that tuning the feature selection threshold “was key to achieving strong results.” I think the paper would be stronger if the authors included analysis of the impact of the threshold in the main text. There is some analysis in the appendix, but it appears that the experiments were only run with a single seed (there is no variance). In addition, it would be interesting to see the tradeoff between feature sparsity and performance (and how this is impacted by the choice of the threshold parameter).
5. **User study has some limitations.** Overall, the user study appears well-executed and provides evidence of the utility of the authors proposed method. However, it does have some notable limitations. The most glaring is that the authors conducted the study on only four test samples from a single dataset. This sample is small and the task is specific, so it’s hard to understand how the findings would generalize beyond the specific cases examined. Further, as the authors acknowledge, it seems like the author’s decisions around how to visualize the explanations produced by each method could impact the results.

[1] Lyu, Qing, Marianna Apidianaki, and Chris Callison-Burch. "Towards faithful model explanation in nlp: A survey." Computational Linguistics (2024): 1-67.

[2] Jain, Sarthak, et al. "Learning to faithfully rationalize by construction." arXiv preprint arXiv:2005.00115 (2020).

[3] Bastings, Jasmijn, Wilker Aziz, and Ivan Titov. "Interpretable neural predictions with differentiable binary variables." arXiv preprint arXiv:1905.08160 (2019).

[4] Yu, Mo, et al. "Rethinking cooperative rationalization: Introspective extraction and complement control." arXiv preprint arXiv:1910.13294 (2019).

[5] Lei, Tao, Regina Barzilay, and Tommi Jaakkola. "Rationalizing neural predictions." arXiv preprint arXiv:1606.04155 (2016).

[6] Jacovi, Alon, and Yoav Goldberg. "Aligning faithful interpretations with their social attribution." Transactions of the Association for Computational Linguistics 9 (2021): 294-310.

[7] Zheng, Yiming, et al. "The irrationality of neural rationale models." arXiv preprint arXiv:2110.07550 (2021).

**Questions:**

My primary concern with this paper is the lack of mention and comparison to existing work on interpretability methods that follow a "select" then "predict" pipeline. Why aren't these mentioned in the paper and why aren't they included as baselines?

---

> ### Author Response · Authors · 2024-11-24
>
> We kindly thank R2 for the careful evaluation of our work and are happy that the reviewer found the writing clarity clear and easy to follow, the experimental analysis thorough over many different datasets, and the explanations useful without sacrificing performance.
>
> We have been able to address each concern in the point-by-point response below and feel that the resulting edits have further improved the manuscript.
>
> **Comparison to explain-then-predict and extractive rationale baselines:** We now provide experiments with a popular extractive rationale, explain-then-predict baseline (FRESH, ACL 2020), compared to InterpretCC FG, InterpretCC GR for both text datasets (SST, AG News), discussed now in Table 2, Sec. 5.1 and Appendix I.4. We tuned the FRESH model over rationale length and learning rate, and ran experiments with 5 random seeds with the exact architecture used in the paper. Notably, FRESH experiments use BERT models, while ICC experiments use the smaller DistillBERT model. For SST, FRESH uses a 30% rationale size with an average accuracy of 82.05% +/- 0.56%, where ICC FG has 88.21% +/- 3.41%, **ICC TopK has 92.98% +/- 0.88%, and ICC GR has 91.75% +/- 1.86%, which are significantly more performant than FRESH**. As SST has 7 words on average, ICC FG selects a larger explanation size, on average 3 or 4 words (indicated in Figure 2), but maintains a higher accuracy. For AG News, FRESH uses a 20% average rational length, with performance of 88.73% +/- 0.69%. This is comparable to the ICC GR numbers 90.35% +/- 1.07% and the the ICC Top K routing approach with 87.25% +/- 2.48%, and more performant than the ICC FG approach 85.72% +/- 5.31% (although the high variation in the FG approach is to be noted), and the 95% CIs overlap showing that the approaches are similar.
>
> In Appendix I.4, we discuss key differences between InterpretCC and extractive rationale methods like FRESH (Jain et al., 2020), summarized as follows: **Task Generalization**: FRESH requires training three separate models sequentially for each task, whereas InterpretCC trains a single model with two parallel parts. **Bias in Saliency**: FRESH relies on different methods to establish initial importance scores (e.g., LIME, attention, gradients), unlike InterpretCC, which learns importances directly. **Model Complexity**: FRESH's 3 models are larger, more complex, and harder to train compared to InterpretCC's simpler architecture. **Explanation Format**: Importantly, FRESH selects contiguous text spans as rationales, suitable for text but not for tabular or time-series data, where InterpretCC maps features to concepts, offering a flexible explanation format for multiple modalities. Due to this limitation, we did not extend the implementation of FRESH to other modality experiments (time-series, tabular) as using contiguous spans of features as explanations would be fundamentally unsuited to the data format.
>
> **A nuanced discussion on the requirement for interpretable features is needed:** We have now added a statement regarding evaluation with interpretable features in Sec. 5, and a discussion of this in Appendix I.5 and Sec. 6. While several of the benchmarks we examined used expert-inspired features (i.e. education datasets), others like the healthcare datasets used unmodified lab data (where measurements still are human interpretable). The labeling of these features does make InterpretCC more useful; the education setting is showcased in the user study. However, we would like to address this concern with a few points:
>
> **A)** For our experiments in the text domain, we did not use human-interpretable features, but worked with the unmodified text. Text, like images, videos, or speech data, is an inherently human interpretable domain (while clickstreams and lab measurements can often be more difficult to interpret). We also saw ICC perform strongly on the OpenXAI synthetic data, where there is no human interpretable meaning.
>
> **B)** It is possible for InterpretCC to use raw time series data, simply specifying the “concepts” as fixed or relative time intervals (analogous to anomaly detection), enabling users to identify which parts of the time series were used in the prediction. Using raw clickstreams directly is rare in human-oriented domains; these domains typically have interpretable features (education, heath, etc.), or can extract interpretations directly (e.g. Asadi et al. AAAI 2023). We now discuss the case for raw time series in Appendix I.5.
>
> **C)** As noted in line 99, we do not address the vision domain, as strong modality-specific interpretability methods already exist (Böhle et al., 2022; You et al., 2023; Donnelly et al., 2022; Thomas et al., 2023). While InterpretCC could be adapted for vision tasks, such as using object detection for grouping, these approaches would be less user-friendly than existing methods. Our architecture specifically targets gaps in other modalities.
>
> [continued in next comment]

---

> ### Author Response · Authors · 2024-11-24
>
> **D)** We envision InterpretCC to work increasingly on LLM extracted features instead of requiring human expert effort. LLMs/LMMs are getting very good at extracting features from raw data (Malberg et al. 2024, Baddour et al. 2024), and as the field moves forward, they will only continue to get better. IntepretCC will be further strengthened by these upcoming advances.
>
> **Explanations produced with InterpretCC can be misleading/unfaithful.** We have now included a discussion (Sec. 6) of the risks of InterpretCC learning correlations that are not intuitive (superficial) in an explanation to make a prediction. By prioritizing human-understandability, our approach could leverage explanations as a tool for auditing models (Fabri et al., 2021; Yadav et al., 2022), with the potential for InterpretCC explanations to reveal a superficial pattern that prompts human expert intervention instead of incorrect model use.
>
> **Limited analysis of impact of sparsity threshold in the main paper.** We have expanded upon the Appendix experiments with additional results on the breast cancer dataset, and moved a discussion into the main paper of the relationship of GS threshold, Tau, and model performance. In Appendix Fig. 6, we had reported the average of several experiment runs (with several seeds). We chose not to show the confidence intervals as it crowds the image and makes it less readable. We are now analyzing significant CIs in Appendix D.2 as well. The revised Fig. 6 additionally includes a new analysis on the relationship between feature sparsity and threshold across both datasets, as well as 8 plots on the relationship between threshold, tau, and model performance. Fig. 13 and 15 granularly show activations across subnetworks for different domains. Although the 95% CIs overlap for all variations, meaning that the performance is not significantly impacted by changing Tau and the GS threshold, the average performance of settings with a high Tau (around 10) and higher thresholds (between 0.7 to 0.8) maintain sparsity, performance, and stability.
>
> **Limitations of user study.** The reviewer comments that the user study was done on only four test samples from a single dataset, potentially limiting the generalizability of the findings.
>
> *Of the 8 datasets, we chose the education domain and the DSP course specifically* because it has been well studied in previous papers (Kidzinsk et al. 2016, Boroujeni et al. 2018, Swamy et al. 2022). The education domain is general enough to allow us to recruit numerous domain experts (in this case, teachers) with a large diversity of profiles and experience, as shown by the socio-economic data we collected, such as their teaching level (see Appendix Fig. 7). This increases the diversity of opinions, hence the generalizability of our findings. Moreover, this dataset offers a concrete and realistic use case where key criteria like actionability and usefulness are easy to assess. The tabular dataset about breast cancer would not allow such diversity in interpretations as it requires niche knowledge, while the synthetic dataset is too abstract for the participants to grasp the pros and cons of each explanation. Finally, the two textual datasets were not suitable for a study, since they are not adapted to the SENN and NAM methods. To our knowledge, this is one of the first user studies done solely on interpretable-by-design models and user preferences, so we wanted to select a setting enabling us to include all approaches.
>
> *Second, we chose not to include more samples due to the length of the study*. Comparing 4 explanations across 5 criteria for 4 samples took nearly 30 minutes for the teachers participating in the study; an excessively long study would decrease the quality of their annotations after several samples. We had experimented with a larger quantity of randomly chosen samples (even up to 10 entries) in 8 pilot experiments, and found strong opinions that it was an extremely mentally taxing task to complete. Therefore, we chose to keep the same samples (chosen randomly) with a more diverse study population.
>
> *Regarding how we chose to visualize the explanations produced by each method*: as we underline in the discussion section, we cautiously post-processed the explanations of SENN and NAM to provide them in a format understandable for a non-technical audience and iterated on the visualizations using a human-centered design process, through extensive iteration with 8 pilot participants. This iterative design resulted in visualization being optimized for teacher perceptions of each method. We note that any imbalance in wording is not necessarily in favor of our method; for instance, users found SENN's explanations more complete than InterpretCC (see Fig. 5).
>
> We feel these suggestions have strongly improved the paper and we hope that our additions have increased confidence in the paper.

---

> > ### Comment · Reviewer_csPd · 2024-11-26
> >
> > I would like to thank the authors for the time and effort spent in answering my review. The authors have addressed all of my comments, and I have updated my score accordingly.

---

> > > ### Author Response · Authors · 2024-11-28
> > >
> > > We thank R2 for reviewing our response, as well as the significant score increase. We are very happy to hear that all concerns have been addressed!

---

### Official Review · Reviewer_xwth · 2024-11-07

**Soundness:** 3
**Presentation:** 3
**Contribution:** 3
**Rating:** 6
**Confidence:** 3

**Summary:**

The paper proposes InterpretCC (ICC), an intrinsic interpretable framework for prediction.

InterpretCC has two modes:

- InterpretCC Feature gating (ICC-FG): here the models rely on a sparse set of different features to make the decision for each instance.

- InterpretCC Group routing (ICC-GR): The feature space is divided into subgroups, subnetworks are trained for each group and interpretCC activates different subnetworks for a given sample.

## Model Architecture:

As described in figure 1, features (for feature gating) or group of features (group routing) are selected via a discriminator network (i.e the discriminator network predicts a mask) the masking is done using Gumbel softmax trick to keep it differentiable.

- For the feature gating: The mask is multiplied by the features and passed to a predictive model to make the final output.

- For the Group routing: The mask is used to select the group of features; each group has a subnetwork, and the final prediction is the weighted sum from the mask and the predictions of the subgroups. This can be viewed as a mixture of expert model where each subnetwork is an expert. Soft masking is used during training to efficiently train the subgroups while hard masking is used during inference. The paper investigates selecting groups in different ways, including handcrafted (user-defined) patterns and using LLMs for grouping.

## Experiments

### Data

The paper showed there approach on Time Series, Text, and Tabular data. For Tabular data the inputs are the features, for text the tokens are the features and for multivariate time series each input across a period of time is a feature i.e they apply the same mask across all time steps. They focused on classification problems.

### Results

- **Accuracy**  ICC was  compared with black box models and interpretable models like SENN and NAM across 8 datasets. Overall, InterpretCC Feature gating seems to outperform interpretable baselines. For non-text datasets ICC can also outperform DNN baselines. For breast cancer dataset, group routing appears to be extremely helpful. They also report that grouping using GPT is, on average, better than other methods, suggesting that using automated grouping methods does not mean compromising performance.

- **Explanations**

  - The paper used the synthetic dataset OpenXAI to evaluate the quality of explanations in comparison to ground-truth, all interpretable methods seem to align to groundtruth explanations in term of faithfulness, while ICC FG ranking of feature importance is higher than others.
  - The paper shows that on all datasets, ICC has high feature sparsity, which makes it more user-friendly.

- **User study**

  -  Four samples were randomly selected (i.e., four students) for prediction from each model, 56 teachers were recruited, the teachers were showed them each model’s prediction of the student’s success or failure along with its explanation. Participants were asked to compare these explanations according to five criteria: usefulness, trustworthiness, actionability, completeness and conciseness. Here SENN, NAM, ICC FR and ICC GR were compared.

  - Overall, ICC models are favored over baselines in 4 out of 5 criteria and in terms of global satisfaction from the domain experts.

**Strengths:**

## Originality -- high
- ICC group routing is original.
- The paper showed that grouping could be done by LLM, and it is as good as the handcrafted ones, showing that this approach can be used on a large scale without requiring extra labor.

##  Quality -- high
- The paper showed ICC can be used on different data types as they tested ICC  on 3 different data domains.
- ICC was benchmarked against strong DNN baselines and good interpretable baselines across 8 datasets.
- For interpretability, OpenXAI was used to show that the explanations produced by ICC match ground truth explanations (although since  ICC produces a mask used to select the input to the model, so this was expected).
- The paper showed that explanations produced by ICC were sparse and, therefore, user-friendly.
- Paper performed a user study that showed domain experts (here teachers) preferred explanations produced by ICC over other interpretable models.
- Overall, the experiment section and the results are very thorough.

## Clarity   -- high
- The paper is well-written and easy to follow.

**Weaknesses:**

## Significance -- Medium
- ICC feature gating is almost the same as SENN feature except with a sparse mask.
- ICC group routing might be inefficient when the number of groups significantly increases, since the model complexity will increase as well.
- ICC sparsity depends on the temperature of the Gumbel Softmax, but its effect was not investigated in the paper.

**Questions:**

-For the group routing, is the input to the discriminator all features (from all groups) and the output just the number of groups? i.e., the discriminator doesn't exactly know which group the features belong to, correct? While each subnetwork only ever sees the features assigned to that subgroup? Please clarify...
- In line 204, you mentioned the following "feature j is activated (the associated value in the mask is non-zero) if the Gumbel Softmax output exceeds a threshold τ , a hyperparameter. This allows the model to adaptively select the number of
features based on each instance, using fewer features for simpler cases and more for complex ones" how was $\tau$ selected?, What is the effect of the changing $\tau$ on the accuracy metrics?
- How are SENN features different from InterpetCC feature gating? Is the only difference that InterpetCC has a Gumbel Softmax while SENN doesn't?

---

> ### Author Response · Authors · 2024-11-24
>
> We kindly thank Reviewer 1 for the careful evaluation of our work and greatly appreciate the reviewer’s positive review of InterpretCC’s high originality, quality, and clarity with “a very thorough experimental section and results”. We strongly agree with all points raised. We have addressed each in a point-by-point response below and feel that the resulting edits have further improved the manuscript.
>
> **ICC feature gating is “almost the same as SENN feature except with a sparse mask”.** We find this an important point to clarify, as there is a significant difference in the underlying design between the two models.
>
> SENN has three components: a concept encoder that transforms the input into a small set of interpretable basis features, an input-dependent parametrizer that generates relevance scores, and an aggregation function that combines them. InterpretCC-FG is a much simplified version of this architecture, where one model is trained end to end with 2 modules instead of 3 models working together. We select important features (via the discriminator network with Gumbel Softmax) and only pass those to the predictive model.
>
> Other than the sparsity and a different training architecture, a key difference is in how the resulting predictive score is obtained in inference. The parametrizer model in SENN Features directly generates scores that are linearly combined for all features. In InterpretCC, the importance selection happens first (reducing the need for extra computation) and the selected features are passed together to a neural network; therefore our predictive module leverages cross-feature interactions in a simpler and more intuitive way, instead of having a score for each feature. We are happy to clarify this in the paper in Sec. 5.
>
> **ICC group routing might be inefficient when the number of groups significantly increases, since the model complexity will increase as well.** Indeed, model size increases with more groups (which require more subnetworks). However the complexity remains the same, as the same discriminator network routing mechanism can simply output a larger vector for a larger number of groups (linear scaling). The human understandability also remains the same (2 groups chosen out of 25 is the same as 2 groups chosen out of 6). Another factor to consider is that with experts specifying the groupings, a large number of groups is less likely to occur.
>
> **Further investigation of the effects of sparsity, specifically on Guble Softmax temperature (“ICC sparsity depends on the temperature of the Gumbel Softmax”).** Our initial submission already includes exploration of this in Appendix Figure 6, exploring the effects of changing Tau and the Threshold parameters on the accuracy for the most popular education dataset. We have now also included additional hyperparameter variation experiments for the healthcare and text cases in Appendix D.2, as well as an extended discussion about this in the main paper.
>
> The updated Appendix Fig. 6 includes a detailed analysis of the effects of modifying Tau and Threshold parameters on the performance of the FG and GR models across both education (time-series) and healthcare (tabular) datasets. We found that while the performance of InterpretCC has overlapping 95% CIs while changing parameters, certain parameter settings have higher variability than others. For both education and health tasks, a Tau of 10 and a Gumbel-Softmax threshold of around 0.7 to 0.8 are performant, sparse in activated features, and relatively stable.
>
> **Use of groupings of features as input for the discriminator network.** The reviewer’s understanding is correct, the discriminator is **not** aware of the feature groups explicitly. However, since the discriminator is trained end-to-end along with the predictive subnetworks, it learns the groupings through the routing logic. We make this more clear in Sec 3.2.

---

> > ### Comment · Reviewer_xwth · 2024-11-25
> > **Thank you**
> >
> > I want to thank the reviewers for the clarifications and for addressing my concerns.
> > My score remains unchanged.

---

> > > ### Author Response · Authors · 2024-11-28
> > >
> > > We thank R1 for reviewing our response. We are grateful to hear that R1's concerns have been addressed.

---

### Author Response · Authors · 2024-11-24
**General Response 1/2**

We are grateful to the chairs and reviewers for their careful evaluation and very happy to receive positive and high-quality reviews.

We agree with all points raised and have been able to reflect and respond to each in detail. We highlight that a major concern of R2 was the lack of comparison to existing NLP interpretability work that follows a “select” then “predict” pipeline. Similarly, R4 asks for a larger discussion of the use of domain knowledge in intrinsic baselines.

We are happy to report that we have implemented an additional baseline comparing our work to popular extractive rationale approaches from the NLP community (Jain et al., ACL 2020) where ICC GR is found to perform on average 5.6% more accurately than FRESH. We also provide an extensive discussion comparing the approaches that R2 and R4 refer to directly in the paper, clearly stating how they methodologically differ from ICC.

REVIEWER 1 provided a very positive review and has recommended acceptance, citing our “original” approach with a “very thorough experiment section and results” described in a “well-written and easy-to-follow” manner. They have requested the following:
1. **A design comparison of ICC Feature Gating to SENN Features.** The models differ not only in ICC's sparseness requirement but also in architecture: SENN uses three components (feature extractor, parametrizer, aggregator), while ICC uses a discriminator and predictor (1 model with 2 parallel parts). Additionally, SENN predicts one score per feature and aggregates them, whereas ICC outputs a single score for the whole model which allows for cross-feature interactions. These details are now included in Sec. 5.
2. **A discussion on the scaling of ICC Group Routing.** Indeed, model size increases with more groups as more subnetworks are involved. However, the complexity (standard routing mechanism) and human-understandability remain the same (e.g. an explanation with 2 groups chosen of 25 is viewed the same as 2 groups chosen of 6).
3. **A further analysis on the effects of changing parameters (e.g. Gumbel Softmax) on the performance of the InterpretCC models.** An initial sparsity and Gumbel Softmax parameter exploration was provided originally in Appendix Fig. 6. We have now conducted additional analyses with a second use case and provided a discussion on this in the paper.
4. **A clarification on the architecture of ICC Group Routing’s discriminator network.** The discriminator does not know the feature groupings as inputs, but as it is trained end-to-end with the predictive module, it learns the groupings through the subnetwork routing.

REVIEWER 2 provided a generally positive review (“simple, intuitive, and novel idea”, “thorough experimental analysis”, “useful explanations without sacrificing performance”, “clearly written and easy to follow”) but recommended a marginal rejection. They cited several reasons motivating this recommendation, which all have been fully addressed:
1. **A comparison to explain-then-predict approaches.** We now provide results of a popular baseline (Sec. 5), as well as a discussion of explain-then-predict approaches in Sec. 2 and Appendix I.4.
2. **A discussion of the requirement of interpretable features for InterpretCC to provide human-friendly explanations.** This is a very important point, and often in real-world situations involving human decision making, features are interpretable at some level (i.e. ranging from the research features used in the education case to the lab measurements extracted without post-processing in the breast cancer case). We discuss extensions regarding raw time series, image, and speech data in the response to R2, and include a nuanced discussion of this in the paper (Sec. 6, App. I.5).
3. **A discussion of the possibility that InterpretCC could use superficial patterns in the selected features for explanation.** This is also a very important point. Referencing Jacovi et al. 2021 and Zheng et al. 2021 as mentioned by R2 as well as Schneider et al. 2021, we discuss the possibility of ICC models learning correlations that are not intuitive (superficial) for explanation in Sec. 6.
4. **Further analysis of sparsity threshold and D.2 experiments**: We expanded parameter sensitivity experiments with new breast cancer data results and added a discussion of this to the main paper. In Fig. 6, we show averages over 5 runs with different seeds (CIs omitted for clarity but now included in the Appendix).

5. **A justification of the small sample of points used in the user study.** In our 8 pilot experiments, we initially started out with a much larger sample of points, and found that teacher attention was greatly reduced after a few evaluations in this intensive study (comparing 4 approaches at once across 5 axes, 4 times). We present this user study as an initial validation of user perception, not a generalizable conclusion.

[continued in “General Response 2/2”]

---

> ### Author Response · Authors · 2024-11-24
> **General Response 2/2**
>
> [continued from “General Response 1/2”]
>
> REVIEWER 3 provided a very positive review stating “the unique strength of this paper is to put user-centric design into intrinsic explanation models”, and praising the decision to “extend models from vision to other modalities”, the “systematic evaluation” with “well-defined metrics”, and the addition of the user study. R3 recommends acceptance with a score of 8 (“accept, good paper”). The reviewer requested the following:
> 1. **A discussion on the interactions of features in the InterpretCC framework, and the possibility to use graph based models.** InterpretCC requests experts (human or LLM) to group features with strong interactions into concepts so the predictive subnetwork can make a decision leveraging the features’ shared insights. Graph models can be very useful here, either adapting the discriminator network with a message passing network like RAINDROP (Zhang et al. 2023) that would output a sparse adjacency matrix as opposed to a sparse vector, or in using graph models as individual subnetworks.
> 2. **A revision of the “if and only if” statement in the introduction.** Great point – indeed, even if InterpretCC selects certain features and these are passed to the prediction module, it is possible (but unlikely) these are ignored in the prediction. We revise the statement as: "The student’s regularity and video watching behavior were the only two aspects selected as important for the student's prediction of passing the course, and the model did not use any other aspects."
> 3. **A discussion on challenging cases for human-oriented feature grouping, and further clarifications on how these feature groups are used in feature gating and group routing.** We discuss this further in the paper in Appendix C. In the difficult case with a feature that belongs equally in two groups, it is better to put a feature in its own subnetwork or combine the groups.
>
> REVIEWER 4 provided a generally positive review, highlighting the “clear and solid motivation of the work” in an “important direction of research”, the “good amount of experiments conducted”, and the “novel” proposed methods) but recommended a marginal rejection. They had several requests motivating this recommendation which have now been fully addressed:
> 1. **A request for a taxonomy and clearer definition of terms in Table 1, and a further discussion of the “faithfulness” and “coverage” of the InterpretCC methods.** We have now clarified definitions with connections to prior work, and added a taxonomy of explainability axes in Appendix A, aligning with the terminology in Agarwal et al. 2022, Pinto et al. 2024, and Swamy et al. 2024.
> 2. **A comparison to intrinsic methods that use expert knowledge.** There are only a few approaches that do this, mostly focused on the vision domain, where concepts are mostly defined through prototype examples (like SENN). We provide a larger comparison to this work in Sec. 2.
> 3. **A discussion on settings where domain knowledge cannot be easily found.** We have now added a nuanced discussion of this for modalities like raw time series.
> 4. **Additional theoretical analysis and intuition of why this MoE approach works.** We believe the specialization of subnetworks correlates to the quality of predictions due to the focus on a few input features. Similarly, adaptive and sparse activation contributes to the reduction of noise. We have now added this to Sec. 6.
>
> To summarize the additional experiments, discussion, and revisions (highlighted in blue in the manuscript), we provide the following results to strengthen our paper:
> - Additional experiments based on explain-then-predict literature in Sec. 5.1.
> - Additional analyses of tuning the sparsity threshold and other parameters, extending the results provided previously in Appendix D.2.
> - A full taxonomy of explainability terms in Appendix A, based in prior literature.
> - An extended discussion on intrinsic methods with concepts/expert knowledge, the requirement of interpretable basis features, potential bias of explanations, and settings where domain knowledge is hard to identify.
> - A theoretical intuition behind the performance of the InterpretCC models in Sec. 6.
>
> Overall, we thank the reviewers for their thoughtful and expert advice that has made us improve the paper much beyond the original submission. We hope that we have adequately addressed reviewer concerns and further improved confidence in our submission.

---

### Meta-Review · Area_Chair_eKSo · 2024-12-16

**Metareview:**

This paper presents a novel intrinsically interpretable Mixture of Experts (MoE) framework that utilizes two neural networks: one for feature gating and the other for group routing. This approach offers interpretability by highlighting the specific features selected from the overall feature set.  The experimental results demonstrate that the proposed method achieves comparable performance to non-interpretable models while providing interpretable outputs.

The paper is well-written and easy to understand. The proposed method exhibits novelty, and the empirical results consistently support the authors' claims.  Importantly, the authors address the crucial challenge of model interpretability.

However, the type of explanation provided – "selected features" – may not always be highly interpretable or actionable. Additionally, the requirement for predefined feature subsets could pose challenges in certain domains.

Despite these limitations, the authors tackle an important problem with a novel approach supported by strong empirical results. Therefore, I recommend accepting this paper.

**Additional Comments On Reviewer Discussion:**

The authors successfully addressed most of the reviewers' concerns during the rebuttal period. Notably, their discussion of the proposed method's novelty compared to existing alternatives led to an increase in their score. Some reviewers raised valid points about the challenges of achieving predefined feature subsets and the limitations of the proposed method's interpretability. Addressing these points in the camera-ready version or future work would further enhance the paper's contribution.

---

### Decision · Program_Chairs · 2025-01-22

Accept (Poster)